# Iceberg calving of Thwaites Glacier, West Antarctica: Full-Stokes modeling combined with linear elastic fracture mechanics

Hongju Yu[1], Eric Rignot[1,2], Mathieu Morlighem[1], and Helene Seroussi[2]

[1]Department of Earth System Science, University of California, Irvine, Irvine, CA, USA
[2]Jet Propulsion Laboratory, California Institute of Technology, Pasadena, CA, USA

*Correspondence to:* Hongju Yu (hongjuy@uci.edu)

**Abstract.** Thwaites Glacier (TG), West Antarctica, has been losing mass and retreating rapidly in the past few decades. Here, we present a study of its calving dynamics combining a two-dimensional flowband Full Stokes (FS) model of its viscous flow with linear elastic fracture mechanics (LEFM) theory to model crevasse propagation and ice fracturing. We compare the results with those obtained with the higher-order (HO) and the shallow-shelf approximation (SSA) models coupled with LEFM. We find that FS/LEFM produces surface and bottom crevasses that are consistent with the distribution of depth and width of surface and bottom crevasses observed by NASA's Operation IceBridge radar depth sounder and laser altimeter, whereas HO/LEFM and SSA/LEFM do not generate crevasses that are consistent with observations. We attribute the difference to the non-hydrostatic condition of ice near the grounding line, which facilitates crevasse formation, and is accounted for by the FS model but not by the HO or SSA models. We find that calving is enhanced when pre-existing surface crevasses are present, when the ice shelf is shortened, or when ice shelf front is undercut. The role of undercutting depends on the time scale of calving events. It is more prominent for glaciers with rapid calving rates than for glaciers with slow calving rates. Glaciers extending into a shorter ice shelf are more vulnerable to calving than glaciers developing a long ice shelf, especially as the ice front retreats close to the grounding line region, which leads to a positive feedback to calving events. We conclude that the FS/LEFM combination yields substantial improvements in capturing the stress field near the grounding line of a glacier for constraining crevasse formation and iceberg calving.

## 1 Introduction

Thwaites Glacier (TG) is the second largest and broadest ice stream in the Amundsen Sea Embayment (ASE) sector of West Antarctica (Fig. 1). Recent observations have reported significant thinning and retreat of this glacier (Rignot, 2001; Shepherd et al., 2002; Pritchard et al., 2009; Rignot et al., 2014). The mass balance of Thwaites was -34±16 Gt/yr in 2007 and this value has been decreasing until present to reach -50 Gt/yr in 2013 (Rignot, 2008; Shepherd et al., 2012; Mouginot et al., 2014). Its grounding line retreated 14 km from 1992 to 2011 (Rignot et al., 2014). The bed elevation of the vast majority of its drainage basin is well below sea level and decreases inland (Tinto and Bell, 2011; Rignot et al., 2014). Such a bed configuration makes the glacier unstable according to the marine ice sheet instability (MISI) theory (Weertman, 1974; Hughes, 1981; Schoof, 2007). With only a small ice shelf able to buttress it, TG may already be in a state of collapse (Parizek et al., 2013; Joughin et al., 2014).

As the glacier retreats farther inland and loses its floating section, its rate of iceberg calving is likely to increase, which would enhance the glacier's contribution to sea level rise (Deconto and Pollard, 2016). It is therefore essential to better understand and simulate the calving dynamics of TG.

Large calving events have been observed on the floating section of TG (Fig. 1b) by satellites (MacGregor et al., 2012). Densely distributed surface and especially bottom crevasses have been revealed by radar depth sounders (Fig. 2). As the buttressing ice shelf calves away and the grounding line retreats, the resistance to flow or buttressing force will decrease, which will favor further retreat and glacier speed up (MacGregor et al., 2012). The calving of icebergs is difficult to model because the processes involved, such as the initiation, propagation and orientation of crevasses, are not well understood and direct observations are rare (Benn et al., 2007). A universal calving law is therefore missing. Most prior studies of crevasse propagation follow the work of Nye (1957), where crevasses propagate based on the balance between longitudinal stress and ice overburden pressure (Bassis and Walker, 2012; Nick et al., 2013; Cook et al., 2014). Although this criterion helps reproduce ice front calving, it does not take into account the stress concentration at the rupture tip of crevasses. This criterion corresponds to the case of multiple closely spaced crevasses (Weertman, 1973; Bassis and Walker, 2012; Ma et al., 2017), but it underestimates the penetration depth of isolated crevasses (van der Veen, 1998b; Plate et al., 2012). To simulate crevasse propagation at the rupture tip, it is necessary to use a fracture theory, such as the linear elastic fracture mechanics (LEFM). This theory has been successfully applied in prior studies to the case of crevasse propagation. van der Veen (1998a, b) used LEFM to model penetration depth of surface and bottom crevasses. Larour et al. (2004a, b) employed LEFM along the rupture tips of ice shelves and showed that the modeled deformation around rupture tips matched observations of ice deformation from fine-scale radar interferometry. Krug et al. (2014) combined LEFM with damage mechanics and reproduced the observed calving front position of Helheim Glacier in Greenland. In their study, however, the crevasse propagation process is not modeled. The crevasses were either zero in size or propagating through the entire ice thickness to create a calving event.

In order to obtain a description of stresses that control crevasse propagation in a time dependent fashion, our approach is to model the viscous flow of the ice using an ice flow model and employ the LEFM theory for crevasse propagation. We apply this approach to study the calving dynamics of TG using the Ice Sheet System Model (ISSM) (Larour et al., 2012). The model simulations are conducted in two dimensions (2D) along a flowline, with geometry based on remote sensing observations. We combine various ice flow models with the LEFM theory to investigate crevasse propagation and iceberg calving. We compare the calving behavior of TG using different initial geometries and different levels of complexity of the numerical ice flow models used to calculate the stress field. We conclude on the importance of using FS for modeling the calving processes of TG and the conditions that are conducive to calving.

## 2   Data and Methods

### 2.1   Data

To model the glacier in 2D, we select a flowline at the center of the fast flowing region of TG as shown in Fig. 1. The flowline is 238 km long, with a 38 km long floating ice tongue (Fig. 3). BEDMAP-2 is used for ice surface, ice bottom and

bed elevation (Fretwell et al., 2013). Over grounded ice, the bed elevation is replaced by the bed elevation computed from a mass conservation method (Morlighem et al., 2011, 2013). At the grounding line, the two datasets display discrepancies in the order of 100 m in a few places, but not along the particular flowline that we selected. The ice temperature field is the steady state temperature computed from the thermal model in ISSM (Larour et al., 2012; Seroussi et al., 2013). The thermal
model is constrained by surface temperature from the regional atmospheric climate model RACMO2 (Lenaerts et al., 2012) and geothermal heat flux from Maule et al. (2005) and includes both conduction and advection processes (Morlighem et al., 2010; Seroussi et al., 2013). The ice surface velocity derived from interferometric synthetic aperture radar (InSAR) data collected in 2008 is used to constrain the ice flow model (Rignot et al., 2011b).

The NASA Airborne Topographic Mapper (ATM) (Krabill, 2014) surface elevation data and the CReSIS MCoRDS ice
thickness data (Gogineni, 2012) provide ice surface and ice shelf bottom elevation, respectively, along flight tracks. We use these observations to evaluate our modeling results. Firn correction is applied to each flight track to ensure that the hydrostatic ice bottom calculated from surface elevation matches the observed ice bottom along the ice shelf. Fig. 2 shows the echograms of two flight tracks along the ice shelf of TG, superimposed by the bed picks from CReSIS, surface from ATM and the hydrostatic ice bottom calculated from these datasets.

## 2.2   Ice Flow Model

The simulations are performed on a 2D flowband model. The basic equations used in our simulations are summarized here for completeness. The ice is considered as an incompressible viscous material driven by gravity. The governing equations of this system are the conservation of momentum and mass:

$$\nabla \cdot \boldsymbol{\sigma} + \rho_i \boldsymbol{g} = \boldsymbol{0} \tag{1}$$

$$\nabla \cdot \boldsymbol{v} = 0 \tag{2}$$

where $\boldsymbol{\sigma}$ is the stress tensor, $\rho_i$ the ice density, $\boldsymbol{g}$ the gravitational acceleration, and $\boldsymbol{v}$ the ice velocity. This governing equation is applied for both the grounded ice and the floating ice. The deformation of ice under stress is described by the constitutive law:

$$\boldsymbol{\sigma}' = 2\mu \dot{\boldsymbol{\varepsilon}} \tag{3}$$

where $\boldsymbol{\sigma}' = \boldsymbol{\sigma} + p\mathbf{I}$, is the deviatoric stress, $p$ the ice pressure, $\mathbf{I}$ the identity matrix, $\mu$ the ice viscosity, and $\dot{\boldsymbol{\varepsilon}}$ the strain rate tensor. The ice viscosity $\mu$ is non-linear and follows Glen's law (Glen, 1955):

$$\mu = \frac{B}{2\dot{\varepsilon}_e^{\frac{n-1}{n}}} \tag{4}$$

where $B$ is the ice viscosity parameter, $\dot{\varepsilon}_e$ the effective strain rate, and $n$ the Glen's law exponent. Here, $B$ is a function of ice temperature with values interpolated from Cuffey and Paterson (2010) and the Glen's law exponent $n$ is set to 3.

For a 2D flowband model, with $(x, z)$ the horizontal and vertical directions, $(u, w)$ the horizontal and vertical velocities, respectively, the above equations can be rewritten as:

$$\frac{\partial}{\partial x}\left(2\mu\frac{\partial u}{\partial x}\right) + \frac{\partial}{\partial z}\left(\mu\frac{\partial u}{\partial z} + \mu\frac{\partial w}{\partial x}\right) - \frac{\partial p}{\partial x} = 0 \tag{5}$$

$$\frac{\partial}{\partial x}\left(\mu\frac{\partial u}{\partial z} + \mu\frac{\partial w}{\partial x}\right) + \frac{\partial}{\partial z}\left(2\mu\frac{\partial w}{\partial z}\right) - \frac{\partial p}{\partial z} - \rho_i g = 0 \tag{6}$$

$$\frac{\partial u}{\partial x} + \frac{\partial w}{\partial z} = 0 \tag{7}$$

This set of equations is the 2D Full-Stokes model and is computationally expensive (Larour et al., 2012). To reduce the computational cost, simplified models may be employed.

There are two widely used simplified models. The first one is the higher-order (HO) model (Blatter, 1995; Pattyn, 2003), which assumes that the horizontal gradient of vertical velocity and the bridging effect are negligible (van der Veen and Whillans, 1989). The governing equations are reduced to:

$$\frac{\partial}{\partial x}\left(4\mu\frac{\partial u}{\partial x}\right) + \frac{\partial}{\partial z}\left(\mu\frac{\partial u}{\partial z}\right) - \rho_i g\frac{\partial s}{\partial x} = 0 \tag{8}$$

where $s$ is the ice surface elevation. The vertical velocity $w$ is decoupled from the system and is computed from incompressibility.

The second model is the Shallow-Shelf Approximation (SSA) model, which makes the additional assumption that the vertical shear is negligible (MacAyeal, 1989). This leads to the following 1D model:

$$\frac{\partial}{\partial x}\left(4H\bar{\mu}\frac{\partial u}{\partial x}\right) - \rho_i g H\frac{\partial s}{\partial x} = 0 \tag{9}$$

where $H$ is the ice thickness and $\bar{\mu}$ the depth-averaged viscosity.

At each time step, the geometry of the flowband is updated by a mass transport model. For FS, the ice surface and ice shelf bottom are treated as two independent free surfaces updated separately:

$$\frac{\partial z_j}{\partial t} + u_j\frac{\partial z_j}{\partial x} - w_j = \dot{m}_j \tag{10}$$

where the subscript $j$ refers to either the ice surface ($j = s$) or the ice shelf bottom ($j = b$) and $\dot{m}_j$ is either the surface mass balance ($j = s$) or the basal melt rate ($j = b$). In HO and SSA, ice surface and bottom elevations are not solved directly. Ice thickness is first solved through a mass transport model:

$$\frac{\partial H}{\partial t} + \nabla \cdot (H\bar{\boldsymbol{v}}) = \dot{m}_s - \dot{m}_b \tag{11}$$

where $\bar{\boldsymbol{v}}$ is the depth-averaged velocity. The surface and bottom elevation of the ice shelf are then updated using hydrostatic equilibrium.

Lateral drag has to be parameterized in a flowband model. Here, it is represented by adding a body force on the ice shelf in the governing equation, as in Gagliardini et al. (2010):

$$f = -\frac{2(n+1)^{\frac{1}{n}}B}{W^{\frac{n+1}{n}}}u^{\frac{1}{n}}; \tag{12}$$

where $W$ is the width of glacier, taken here as 130 km. The convergence of ice from upstream to downstream also needs to be taken into account to conserve mass. Here, we first calculate the ice mass flux along the flowline. Then, we add an artificial surface mass balance term, $\dot{m}_a$, to the original surface mass balance, $\dot{m}_s$, to ensure that the ice mass flux is constant from the inflow boundary to the grounding line.

## 2.3 Boundary Conditions

At the ice surface, the atmospheric pressure exerted on ice is negligible and thus a stress free boundary condition is applied:

$$\boldsymbol{\sigma} \cdot \mathbf{n} = 0 \tag{13}$$

where $\mathbf{n}$ is the unit normal vector pointing outward.

At the bed, boundary conditions are different for grounded ice and floating ice. For grounded ice, the basal drag is assumed to follow a linear friction law:

$$\boldsymbol{\tau_b} = -\alpha^2 \boldsymbol{v}_b \tag{14}$$

where $\boldsymbol{\tau_b}$ is the basal drag, $\boldsymbol{v}_b$ the velocity tangential to the bed, and $\alpha$ the friction coefficient. Here, $\alpha$ is inferred from an inversion so that the modeled surface velocity matches the observed surface velocity (Section 2.4). Other sliding laws have been proposed in the past, including a non-linear friction law (Weertman, 1957) and a friction law that includes effective pressure at the bed (Budd et al., 1979). Here, the simulation time is short, the grounding line does not migrate, and the changes in ice thickness are small. The impact of the sliding law is therefore limited and we choose to use a linear sliding law for simplicity.

At the ice shelf bottom and the ice front, seawater pressure is applied at the ice-ocean boundary:

$$\boldsymbol{\sigma} \cdot \mathbf{n} = 0 \qquad z \geq 0 \tag{15}$$
$$\boldsymbol{\sigma} \cdot \mathbf{n} = \rho_w g z \, \mathbf{n} \qquad z < 0 \tag{16}$$

where $\rho_w$ is the seawater density and sea level is at $z = 0$. In our simulations, the ice shelf bottom elevation, $z_b(t)$, is unknown when applying this boundary condition. A replacement with $z_b(t - dt)$, with $dt$ the time step, produces large vertical velocities that destabilize the system (Durand et al., 2009a). A shelf dampening term based on ice velocity and geometry is therefore added to $z_b(t - dt)$ to approximate $z_b(t)$:

$$z_b(t) = z_b(t - dt) + \boldsymbol{v} \cdot \mathbf{n} \sqrt{1 + (\partial z_b(t - dt)/\partial x)^2} dt \tag{17}$$

For SSA and HO, $z_b$ is known because we solve for ice thickness and use hydrostatic equilibrium to calculate the ice surface and bottom elevation. The dampening term is therefore not required.

The grounding line position is computed at every time step. For FS, it is treated as a contact problem (Nowicki and Wingham, 2008; Durand et al., 2009b; Drouet et al., 2013). At the ice-bedrock-ocean boundary, the grounding line will retreat if the water

pressure is higher than the normal stress exerted by the ice. At the ice-ocean boundary, a non-penetration condition is imposed. For HO and SSA, the migration of the grounding line is determined by the hydrostatic equilibrium (Seroussi et al., 2014). At the inflow boundary, a Dirichlet boundary condition is applied for the velocity. The horizontal velocity is taken from InSAR-derived ice velocity data (Rignot et al., 2011b) and the vertical velocity is set to 0.

## 2.4  Inversion for Basal Friction

We have no direct observation of basal friction. In order to obtain a realistic representation of the basal conditions, we use an adjoint method as in Morlighem et al. (2010, 2013) to find a distribution of the basal friction coefficient, $\alpha$, that minimizes a cost function:

$$\mathcal{J}(u,\alpha) = \int_{\Gamma_s} c_1 \frac{1}{2} (u - u_{obs})^2 d\Gamma + \int_{\Gamma_s} c_2 \frac{1}{2} \ln\left(\frac{|u|+\epsilon}{|u_{obs}|+\epsilon}\right)^2 d\Gamma + \int_{\Gamma_b} c_3 \frac{1}{2} \left(\frac{\partial\alpha}{\partial x}\right)^2 d\Gamma \tag{18}$$

where $u$ is the modeled surface velocity, $u_{obs}$ the observed surface velocity, $\epsilon$ a minimum value ($10^{-8}$ m/yr) to avoid zero velocity, $\Gamma_s$ and $\Gamma_b$ the ice surface and bedrock, respectively. The first term of this cost function represents the misfit between modeled and observed velocity. The second term allows a better representation for slow flow regions and the third term is a Tikhonov regularization term, invoked to avoid unphysical short scale spatial variations of $\alpha$ (Vogel, 2002). We calibrate $c_1$ and $c_2$ so that the first and second terms have the same order of magnitude and we calibrate $c_3$ using an L-curve analysis approach (Hansen, 2000).

## 2.5  Linear Elastic Fracture Mechanics Model

A physically-based LEFM model is used to simulate crevasse propagation. In the LEFM theory, there are three modes to open a crevasse: mode I opening, mode II sliding and mode III tearing (Anderson, 2005). Only mode I is considered here. The key variables in LEFM are the stress intensity factor $K(x, z, t)$ and the fracture toughness $K_c$. If $K$ is larger than $K_c$, a crevasse will propagate. For a crevasse at a given location with a given stress field, $K$ is computed through the integration of the normal stress from the bottom of the crevasse to the tip of the crevasse (van der Veen, 1998b). For bottom crevasses, the equations are:

$$K = \int_b^{b+h} \frac{2\sigma_n(z)}{\sqrt{\pi h}} G(z, h, H) dz \tag{19}$$

$$\sigma_n(z) = \sigma'_{xx}(z) + \rho_w g z - \rho_i g(s - z) \tag{20}$$

where $h$ is the height between the tip and the bottom of the crevasse, $b$ the elevation of the ice shelf bottom, $H$ the ice thickness, $\sigma'_{xx} = \sigma_{xx} + p$ the deviatoric stress, with $\sigma_{xx}$ the longitudinal stress and $p$ the pressure and $G$ a weighting function (Krug et al., 2014). For surface crevasses, the equations are similar with the water pressure term equal to zero since we assume no melt water production at the surface. $K_c$ is a material property and previous studies showed that $K_c$ ranges from 0.1 to 0.4 MPa m$^{1/2}$ for ice (Fischer et al., 1995; Rist et al., 1996, 2002). Here, $K_c$ is set to 0.2 MPa m$^{1/2}$ following Krug et al. (2014).

A simple algorithm for the combination of ISSM and LEFM is described in Fig. 4. First, a position is chosen arbitrarily as the initial crevasse position. ISSM is used to calculate the stress field. With this stress field at the location of the initial crevasse, the LEFM theory is used to find the maximum heights of the surface and bottom crevasses that satisfy $K > K_c$. However, this criterion is never satisfied when the crevasse depth is small (cm scale) and a minimum depth is required. Here, we assume that a crevasse can propagate if its minimum required depth is smaller than 1 m. This process is assumed to be instant and the stress field is assumed to be unchanged (Duddu et al., 2013; Ma et al., 2017). Once the crevasse is opened, its width is assumed to grow to 20 m instantaneously (our mesh resolution is 5 m). The geometry is then updated to include the new crevasses. Numerically, this is done by migrating each node vertically, but none of the nodes is removed from the mesh. The new ice geometry is allowed to adjust viscously with ISSM for a period of 0.01 yr during which the crevasse becomes wider, shallower, and smoother due to the viscous deformation of ice. The computed pressure also becomes close to the hydrostatic pressure during this period as the singularity in the pressure field near the crevasse tip is diminished. A series of tests conducted with shorter time steps do not indicate any change in the results (Fig. S1). When the shape of a crevasse is adjusted viscously, its width violates LEFM assumptions. The pre-existing crevasse is therefore considered as a feature on the ice shelf and affects the stress field computed from the viscous model. When the LEFM is called again, it is applied to an infinitesimal crevasse at the apex of the pre-existing crevasse. The new crevasse, if it propagates, grows to 20 m wide instantly and then merges into the pre-existing crevasse through viscous deformation. Calving is assumed to occur when the surface crevasse reaches sea level or when the bottom crevasse reaches the ice surface (Benn et al., 2007).

The limitations of our approach are as follows. The LEFM approach does not explain the propagation of a crevasse from 1 mm to 1 m scale (Weiss, 2004). Therefore, we assume that a crevasse can propagate as long as its minimum required depth is smaller than 1 m. The initiation of crevasses could be improved using a subcritical crevasse propagation method or damage mechanics (Weiss, 2004; Krug et al., 2014), but this is beyond the scope of our study. Another issue is associated with the width of crevasses, which should be ∼1 cm according to LEFM (Lister, 1990; Bassis and Ma, 2015). Modeling a crevasse at this scale is computationally too demanding. Once a crevasse is formed, however, its shape is controlled by the viscous flow of ice, which reduces its depth and increases its width. In our experiments, crevasses are able to grow quickly from 20 m width to 60-70 m in 0.01 yr when deforming under viscous flow. This viscous widening process is not sensitive to the width itself like elastic widening. Experiments with an initial width of 10 m and 40 m are conducted and the crevasses width are similar at the end of these experiments. We therefore deem it reasonable to assume that the crevasses grow to a width of 20 m if LEFM shows that the infinitesimal crevasse can propagate.

## 3   Simulations

### 3.1   FS model validation

ISSM is a coupled, thermo-mechanical, finite element, ice flow model (Larour et al., 2012). The three models, FS, HO and SSA are implemented in ISSM, which makes it practical to compare their performance (Morlighem et al., 2010; Seroussi et al., 2011). To validate our FS modeling of the grounding line dynamics, we run the Experiment 3 of MISMIP. In this experiment,

we model the grounding line migration resulting from a change in ice rheology on an over-deepened bed (Pattyn et al., 2012). The results, shown in Fig. 5, indicate that the grounding line is unstable on a retrograde bed and displays a hysteresis behavior in response to perturbations in ice rheology. This is consistent with the MISI theory, the analytical solution and other numerical models (Weertman, 1974; Schoof, 2007; Pattyn et al., 2012). The steady state grounding line positions obtained by ISSM agree with the FS solution obtained by Elmer/Ice (Durand et al., 2009a), to within 15 km. The results are also in good agreement with the analytical solution of Schoof (2007), especially in the retreating phase (step 7-13), to within 20 km. In the advancing phase, the difference is larger, ∼50 km. However, this level of discrepancy in grounding line position is considered to be satisfactory and has been attributed to numerical issues associated with mesh resolution (Durand et al., 2009a; Pattyn et al., 2012). We therefore conclude that ISSM is able to reproduce the results of MISMIP Exp 3.

## 3.2    Model Setup

In our simulations, the horizontal resolution of the mesh is 100 m, refined to 5 m within 3 km of the initial crevasse position. Vertically, the domain is uniformly discretized into 40 layers. In total, the domain has 281,680 elements. The time step we choose is 0.0005 yr (∼4.4 hr) and the LEFM model is called every 0.01 yr. The simulations are run for 0.3 yr or until calving occurs, whichever happens first. In all following experiments, the basal melt rate is chosen so that the grounding line does not migrate and the ice shelf bottom has a stable elevation (within few meters).

Five sets of experiments, labeled Exp. A–E, are conducted to simulate the propagation of crevasses. In the first set, eleven experiments, Exp. A1–A11, are run with infinitesimal initial crevasses, zero crevasse depth and width, at both the surface and the bottom. In these experiments, the numbers 1–7 indicate crevasses initiated near the grounding line (at distances x = 0.5, 1, 1.5, 2, 2.5, 3, 3.5 km downstream of the grounding line); the numbers 8 and 9 indicate crevasses initiated in the middle of the ice shelf (x= 18, 28 km); and the numbers 10 and 11 indicate crevasses near the ice front (x = 35, 36 km). The initial crevasse positions are chosen to be more densely spaced in the grounding line region as the stress conditions in this region are more complicated and exhibit more spatial variations.

In the next four sets of experiments, the initial glacier geometry is altered to evaluate its impact on crevasse propagation. The second (Exp. B1–B7) and the third (Exp. C1–C3) sets are designed to test the stability of TG with a shortened ice shelf. The length of the ice shelf is reduced from 38 to 4 km (Exp. B) and 2 km (Exp. C), respectively. In the fourth set of experiments (Exp. D1–D7), a 3 m deep, 100 m wide initial surface crevasse is added to the initial geometry while the initial bottom crevasse is still kept as a infinitesimal crevasse. In the last set (Exp. E1–E7), we undercut the ice shelf front of a 4 km–long ice shelf by 400 m over the last 400 m. The initial crevasse positions for experiments B-E are the same as Exp. A.

## 4 Results

### 4.1 Inversion

The inversion results of FS, HO and SSA are shown in Fig. 6. For all three models, the inferred basal friction coefficient, $\alpha$, has similar values and spatial patterns. The modeled ice surface velocities are in reasonable agreement. The modeled surface velocity after inversion closely matches the observed surface velocity over grounded ice. However, there remains a 200 m/yr, or 6%, difference in the grounding line region and on the ice shelf. We attribute this discrepancy to errors in ice rheology and uncertainties associated with the parameterization of the lateral drag.

### 4.2 Observed crevasses

In the data acquired by NASA ATM and CReSIS MCoRDS from 2009 to 2014 (Gogineni, 2012; Krabill, 2014), we find that surface and bottom crevasses are densely distributed on the ice shelf of TG (Fig. 1b and Fig. 2). With these data, we estimate the height and width of each surface and bottom crevasse (crevasses narrower than 200 m are neglected because of the high uncertainties in their depth and width). The results are shown in Fig. 7. We find that the mean height is 18.7 m for surface crevasses and 103.1 m for bottom crevasses. The height of surface crevasse ranges from 2–82 m, but 90 % of them are within 2–40 m. The height of bottom crevasses ranges from 20–270 m. The mean width for surface and bottom crevasses are 821 m and 724 m, respectively, and 80 % of the crevasses have a width ranging from 300 m to 1000 m. The nominal measurement error is 10 cm for the ATM-derived ice surface elevation. However, at some data points, especially on the ice shelf, the error can reach a few meters (Krabill, 2014). The measurement error for the MCoRDS-derived ice bottom elevation is 14 m (Gogineni, 2012).

### 4.3 Deviation from hydrostatic equilibrium

In the grounding line region, i.e. within 5–10 km downstream of the grounding line, ice is pushed down below hydrostatic equilibrium because of a bending moment applied on the ice that arises as the basal regime changes abruptly across the grounding line. In TG, the ice is tens of meters below hydrostatic equilibrium (Fretwell et al., 2013). In our selected flowline, the maximum deviation is 85 m. In the two flight tracks shown in Fig. 2, we find a maximum deviation of 130 m for track PQ and 122 m for track RS. In addition, in the region where surface and bottom crevasses are present, the deviation is larger and measured in hundreds of meters (Fig. 2). In the FS solution, it is possible to account for this non-hydrostatic condition. For instance, we obtain a maximum deviation of 68 m in a steady state solution for our selected flowline.

### 4.4 Crevasse propagation

The evolution of K for selected experiments with different models is shown in Fig. 8. For HO and SSA, the crevasses do not propagate if the initial crevasse position is >2000 m downstream of the grounding line. When the crevasse propagates, the stress intensity factor decreases. The crevasse then stops growing when $K < K_c$ and closes up due to the viscous deformation.

At the end of the simulations, the bottom crevasse never exceeds 50 m, which is small compared to observations (Fig. 7). In other words, under the assumption of hydrostatic equilibrium, which is required by HO and SSA, the crevasses cannot grow and generate calving events when combined with the LEFM theory. In the remainder of the study, we therefore only discuss the FS case.

In the first set of experiments (Exp. A1–A11), with the initial geometry and infinitesimal crevasses on the top and the bottom of the ice shelf, the crevasses of all eleven cases stop growing at the end of the simulations and none produce a calving event (Fig. 9). The final height of bottom crevasses is 200–300 m near the grounding line (Exp. A1–A7) and 50–100 m downstream (Exp. A8–A11). The surface crevasses are one order of magnitude smaller, 10–15 m near the grounding line and 2–5 m downstream. The width of all crevasses is between 400 and 500 m.

The results of the experiments with varying initial geometries are shown in Fig. 10. With an ice shelf shortened to 4 km, calving occurs within 1 km of the ice front (Exp. B6 and B7, Fig. 10a) and the other experiments (Exp. B1–B5) have results similar to the initial 38 km long ice shelf (Exp. A1–A5), i.e. the final bottom crevasse height does not exceed 200–300 m. With an ice shelf shortened to 2 km, calving occurs in all three experiments (Exp. C1–C3, Fig. 10b).

In Exp. D1–D7, where we add a 3 m deep, 100 m wide, initial surface crevasse, calving occurs for crevasses located within 1.5 km of the grounding line (Exp. D1–D3, Fig. 10c). Further downstream (Exp. D4–D7), the crevasse propagation is identical to the case with infinitesimal surface crevasses (Exp. A4–A7).

In the last set, where the ice shelf is shortened and undercut, we find that calving occurs within 1.5 km of the ice front (Exp. E5–E7, Fig. 10d). In regions where calving does not occur, undercutting vanishes slowly within 0.1 yr due to the viscous deformation and downstream advection of ice.

Among all experiments, only Exp. B6 produces calving caused by a surface crevasse propagating to sea level and it takes 0.24 yr for the calving to occur. For all other calving cases, calving occurs because a bottom crevasse propagates to sea level and the process is five times more rapid, i.e. within 0.05 yr. For the cases that calving does not take place, the crevasses stop growing before the end of the simulations. In Exp. A5–A11, the modeled crevasses undergo a non-monotonic evolution where the crevasses depth decreases in a few time steps during the simulation (Fig. 9). This evolution is caused by the temporal change in K. K decreases when the crevasse propagates until it stops growing. K then increases when the crevasse become shallower viscously until it can propagate again.

## 5 Discussion

The width and depth of surface and bottom crevasses produced by our crevasse propagation experiments with FS are within the range of the width and depth of surface and bottom crevasses observed by ice radar sounders (Fig. 7). This suggests that the combination of FS with the LEFM theory is a realistic way to model crevasse propagation and iceberg calving. Some observed crevasses are wider than our modeled crevasses because the width of crevasse is still increasing at the end of our simulations (the depth is stable). Furthermore, we do not include ocean forcing in our model, which could affect crevasse growth. There are also some observed surface crevasses that are much deeper than our modeled crevasse. However, a number of these surface

crevasses correspond to rift, i.e. ice surface is close to sea level, hence nearly equivalent to calving events. With HO and SSA, however, because of the assumption of hydrostatic equilibrium, the water pressure term and the overburden ice pressure term in Eq. (20) cancel each other at the bottom of the ice shelf and thus the bottom crevasses are unable to grow to a size that matches observations. With the non-hydrostatic condition included, the two pressure terms in Eq. (20) do not cancel each other

with FS in the region near the grounding line or the region with crevasses, which helps propagate the crevasses. In the radar echograms, large bottom crevasses (over 100 m) are also observed along the ice shelf, tens of kilometers downstream of the grounding line. According to our results from Exp. A, the crevasses formed in the grounding line region stop growing once they reach a stable size. Therefore, we posit that these crevasses are the result of advection of crevasses formed upstream. In summary, the non-hydrostatic condition plays a major role in crevasse formation. Not accounting for this condition makes it

difficult to explain the observed crevasse pattern.

In our simulations, we find that crevasses propagate significantly faster near the ice front when the ice shelf is shortened. In principle, the length of a nearly non-confined ice shelf, such as the floating ice tongue of TG, should not have a major impact on the buttressing that the ice shelf exerts on grounded ice. Here, we find that the propagation of crevasses near the ice front, while limited for the initial 38 km-long ice shelf, becomes significantly enhanced with a shortened ice shelf. When the ice shelf

is shortened, the longitudinal stress near the ice front increases at the surface and decreases at the bottom. The increase in the surface stress makes it easier for the surface crevasse to propagate, while the decrease in bottom stress prevents the propagation of the bottom crevasse. Over time, the stress at the bottom increases and the surface crevasse grows. The bottom crevasse is then able to propagate quickly through the entire ice column to cause calving because of the large difference between the water pressure and the overburden ice pressure. If calving takes place and creates a shorter ice shelf, our model predicts that the new

ice shelf will be more prone to calving, i.e. a positive feedback.

When an initial crevasse of 3 m depth and 100 m width is added to the surface, we find that the surface crevasse grows quickly to 35 m before the bottom crevasse starts to propagate. The large difference between the water pressure and the overburden ice pressure at the bottom however, makes the bottom crevasse propagate rapidly through the entire ice thickness and produces calving. This is consistent with Bassis and Walker (2012), who suggested that ice shelves are difficult to form in the presence

of pre-existing crevasses. However, long ice shelves calving at the grounding line region is not something commonly observed on TG. Three reasons might explain this result. One reason is that we assume that a surface crevasse aligns perfectly with a infinitesimal crevasse at the bottom, which is not certain. A second one is that bottom crevasses could also form from thermal cracking (Humbert and Steinhage, 2011; Vaughan et al., 2012), in particular not aligned with a surface crevasse. Thermal cracking would facilitate the propagation of a bottom crevasse. If the corresponding surface crevasse remains shallow, the

seawater-filled bottom crevasse formed by thermal cracking will not propagate far because the difference between the water pressure and the overburden ice pressure will be smaller than in the presence of a deep surface crevasse. The third reason is that most surface crevasses are formed in train, whereas here we only model one. A train of crevasses creates a shielding effect, which effectively reduces the stress concentration at rupture tips and anneals the propagation of crevasses (van der Veen, 1998b; Krug et al., 2014).

Undercutting on the ice front is a common feature, especially for tidewater glaciers with a short to non-existent floating section (Rignot et al., 2010). In a prior study, O'Leary and Christoffersen (2013) suggested that undercutting leads to significant changes in the stress field that enhances calving. Cook et al. (2014) argued, however, that the change in stress field is only significant in diagnostic simulations and is much smaller in prognostic simulations. Krug et al. (2015) also showed that undercutting has no effect on the glacier mass balance on annual time scales. Here, we find that undercutting does affect the stress field significantly near the ice front but its impact on calving depends on the time scale of calving events. Undercutting increases the surface stress and decreases the bottom stress just as in the case of a shorter ice shelf and thus induces a similar type of calving. The influence of the stress field is however time dependent due to the viscous adjustment of ice. In our simulations, we find that if the undercutting is not large enough to produce calving within about 0.1 yr, then it will have no impact on calving. If calving occurs on shorter time scales, then undercutting significantly enhances the process. The high melt that produces undercutting, however, is not considered in our simulation. If the high melt is sustained, which depends on the seasonal variability of thermal forcing from the ocean, the time scale of undercutting will be longer. This conclusion may help reconcile the previous studies because it shows that the impact of undercutting depends on the time scale of calving events. We conclude that the impact of undercutting will be more significant for fast-moving glaciers with high calving rates than for slow moving glaciers with a low calving rate. A high calving rate will give less time for the glacier to adjust viscously to the undercutting than for a slow calving glacier. As a conjecture, since glaciers with a high calving rate have more impact on the total mass balance, we conclude that undercutting is an important factor in the study of calving dynamics.

In this study, the simulations are conducted in a 2D flowband model with one crevasse propagation event. It would be of interest to generalize the present simulation to a 3D geometry with the inclusion of multiple crevasses and a moving ice front. In 3D, a better representation of the lateral shear and a complete surface/bed geometry will provide a more realistic context for the models. The simulation of a series of calving events with a train of crevasses over a long time period would provide more realistic information about how a glacier will respond to a calving event in terms of the migration of its grounding line and the evolution of its ice speed. It would also be useful to include in the current model the subcritical propagation or the damage mechanics, which is needed to improve the simulation of the initial propagation from an infinitesimal crevasse to a ∼1 m deep, ∼10 m wide crevasse (Weiss, 2004; Krug et al., 2014). With these additions, we would be in a better position to project calving events.

## 6   Conclusions

We use a two-dimensional flowband Full-Stokes model coupled with LEFM theory to model the calving behavior of Thwaites Glacier. We find that FS combined with LEFM produces crevasses consistent in width and depth with observations and produces calving events, whereas the HO and SSA models do not. The reason for the propagation of crevasses is the existence of a non-hydrostatic condition of ice immediately downstream of the grounding line, which is not accounted for in simplified models that assume hydrostatic equilibrium everywhere on the ice shelf. We also find that calving is enhanced in the presence of pre-existing surface crevasses, shorter ice shelves, or if the ice front is undercut. We conclude that it is important to consider the full stress regime of ice in the grounding line region to replicate the conditions conducive to calving events, especially the

non-hydrostatic condition that is critical to propagate the crevasses. Further studies ought to examine how these results vary in 3D and including the role of lateral margins.

*Acknowledgements.* This work was carried out at the University of California Irvine and at California Institute of Technology's Jet Propulsion Laboratory under a contract with the Cryosphere Science Program of the National Aeronautics and Space Administration, grant NNX14AN03G. We thank the reviewers T.Wagner, J. Bassis and J. Todd for their insightful and helpful comments.

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

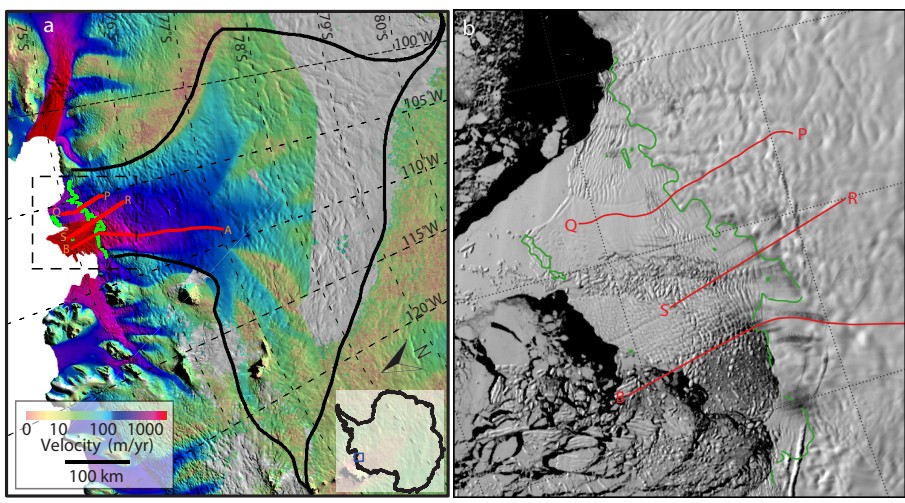

**Figure 1.** Velocity map and MODIS image of Thwaites Glacier (TG), West Antarctica. a) Velocity field of TG derived from InSAR with data collected in 2008 (Rignot et al., 2011b). The black contour is the drainage basin of TG. b) MODIS image of the dashed box region in a) on Nov. 01, 2012. PQ and RS are the flight tracks of the echograms shown in Fig. 2. AB is the selected flowline of this study. The green line is the grounding line of TG in 2011 (Rignot et al., 2011a).

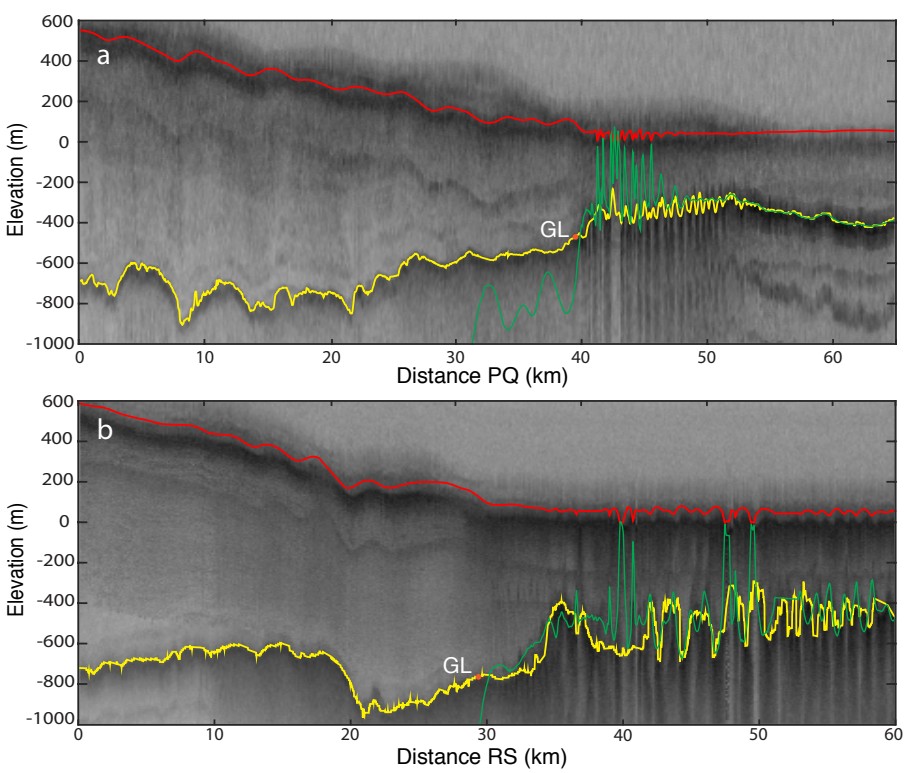

**Figure 2.** Two echograms of Thwaites Glacier (TG). a) Echogram of flight track PQ on Nov.02, 2009. b) Echogram of flight track RS on Nov.19, 2010. The red lines are ice surface elevation measured by Airborne Topographic Mapper (ATM) (Krabill, 2014) and the green lines are bed elevation calculated from hydrostatic equilibrium. The yellow lines are the elevation of ice bottom measured by ice radar depth sounder (Gogineni, 2012). The orange dots are the grounding line positions in 2011 (Rignot et al., 2011a).

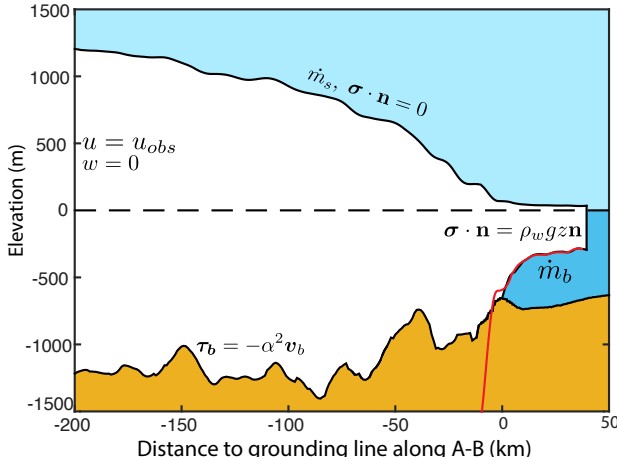

**Figure 3.** Geometry of the selected flowline AB and boundary conditions of the model. The black lines are ice surface elevation, ice bottom elevation and bed elevation. The red line is the hydrostatic bottom elevation calculated from surface elevation.

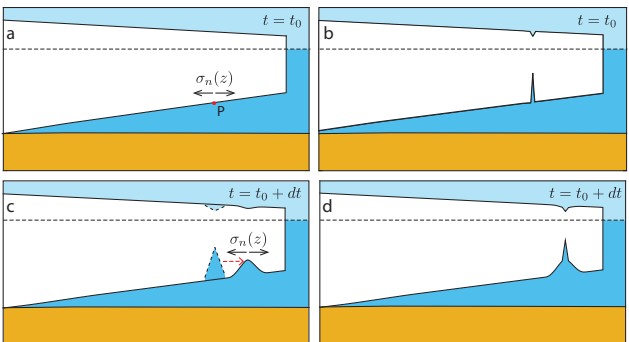

**Figure 4.** Schematic of the combination of ISSM and LEFM. a) Initial condition, b) Crevasses propagate, c) Crevasses advect downstream, d) Crevasses grow.

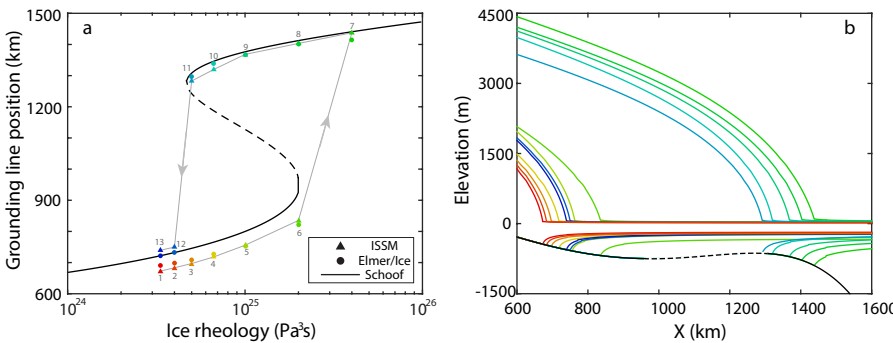

**Figure 5.** Results of MISMIP Exp 3. a) Steady state grounding line positions of MISMIP Exp 3. Triangles are results of ISSM; circles are results of the FS solution of Elmer/Ice Durand et al. (2009a) and the black curve is Schoof (2007) solution (Pattyn et al., 2012). The gray arrow shows the sequence of ice rheology perturbation at each step. b) Steady state profile at each step obtained by ISSM. The retrograde part of the bed is shown in dashed line.

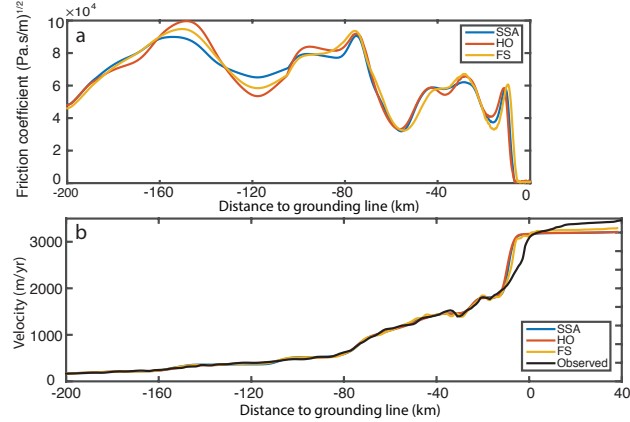

**Figure 6.** Inversion results of basal friction on flowline AB. a) Friction coefficient inferred with all three models (FS, HO and SSA). b) Comparison of modeled surface velocity and observed surface velocity for all three models.

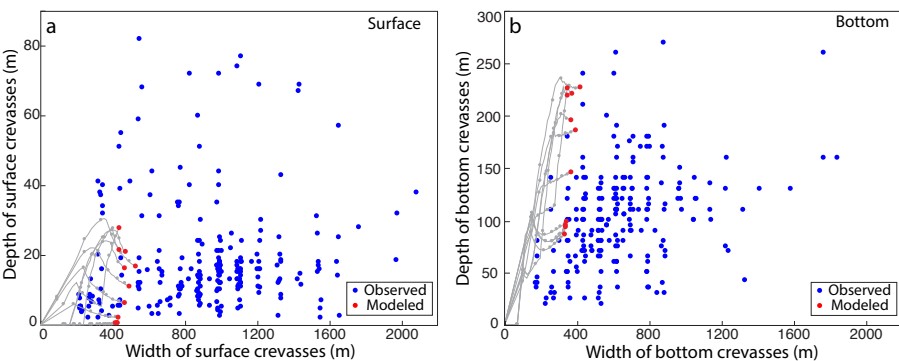

**Figure 7.** Comparison of the shape of observed and modeled crevasses. a) Depth and width of surface crevasses. b) Depth and width of bottom crevasse. Blue dots are observed crevasses and red dots are modeled crevasses from Exp. A. Gray lines and gray dots are the evolution of the shape of modeled crevasses.

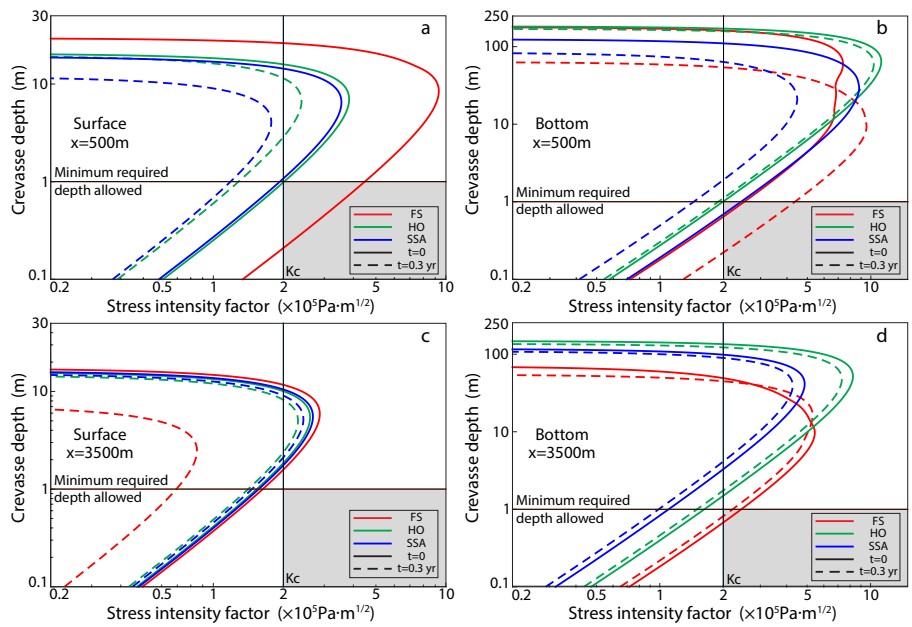

**Figure 8.** Stress intensity factor as a function of depth. a) Surface crevasse with initial crevasse position at x=500 m. b) Bottom crevasse with initial crevasse position at x=500 m. c) Surface crevasse with initial crevasse position at x=3500 m. d) Bottom crevasse with initial crevasse position at x=3500 m. Red, green and blue lines are corresponding to the FS, HO and SSA model. Solid and dashed lines are corresponding to the beginning and the end of each simulation. (The stress intensity factor for surface crevasse of FS at x=500 m, t=0.3 yr is not shown because it is negative at all depth.) The crevasse propagates if its minimum required depth is smaller than 1 m, (i.e., if the curve passes through the fourth quadrant).

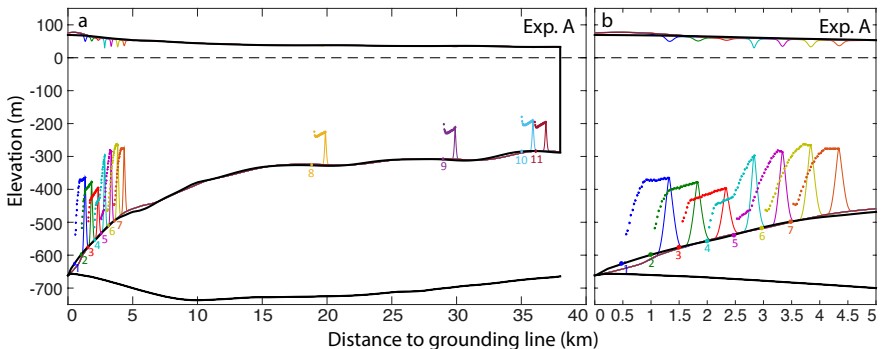

**Figure 9.** Crevasse propagation with the initial geometry of flowline AB. a) Crevasse propagation of Exp. A1–A11 with FS. Each color corresponds to one initial crevasse position, indicated by the number. The solid lines are the shape of final crevasses. The dotted lines are the evolution of the tips of bottom crevasses. b) Details of the grounding line region for Exp. A1-A7. The black lines are the initial geometry for ice surface, ice bottom and seafloor.

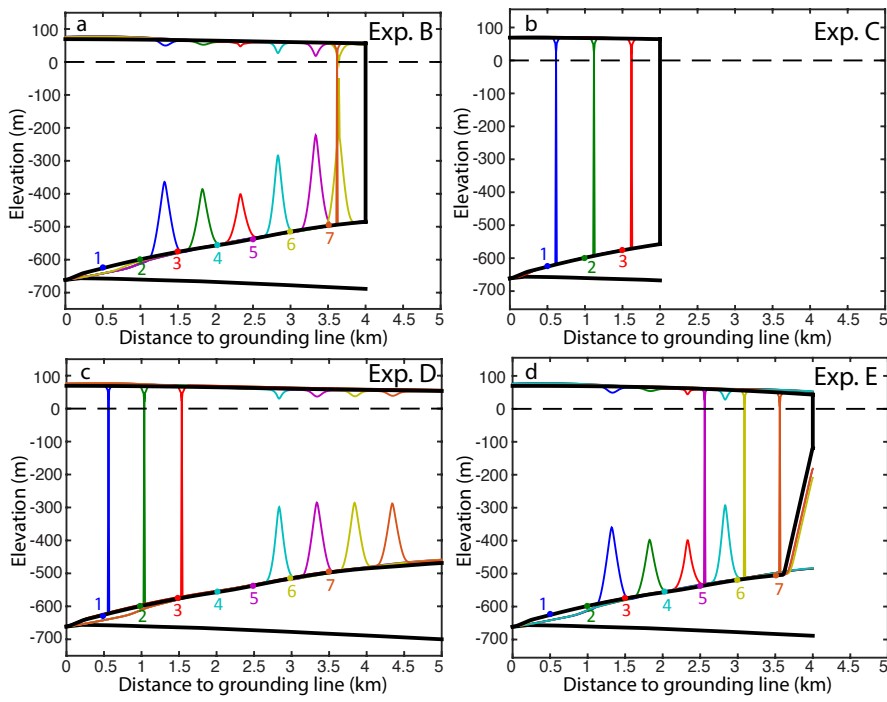

**Figure 10.** Crevasse propagation in the grounding line region with varying initial geometry. In each panel, solid lines are the shape of final crevasses with a) 4 km long ice shelf (Exp. B1–B7), b) 2 km long ice shelf (Exp. C1–C3), c) 3 m deep, 100 m wide initial surface crevasse (Exp. D1–D7), and d) 4 km long ice shelf with a 400 m wide and 400 m high undercut ice front (Exp. E1–E7). The black lines are the initial geometry for ice surface, ice bottom and seafloor.