# Peer review of "Iceberg calving of Thwaites Glacier, West Antarctica: Full-Stokes modeling combined with linear elastic fracture mechanics"

_The Cryosphere, 2016_

## Referee Comment (RC1) · T. Wagner (Referee) · 14 Dec 2016

**Review of** *"Iceberg calving of Thwaites Glacier, West Antarctica: Full-Stokes modeling combined with linear elastic fracture mechanics"* **by Yu et al.,**

Review by Till Wagner (tjwagner@ucsd.edu)

The authors develop a 2D ice-sheet model that includes the time-evolution of crevasses, by combining a Full Stokes (FS) model with an account of linear elastic fracture mechanics (LEFM). The results are compared to those obtained with higher-order (HO) and shallow-shelf approximations (SSA). The model is set up to represent a cross-section of Thwaites Glacier (TG), and the model output is compared to observed crevasses at TG. The results suggest that deviations from hydrostatic equilibrium are an important driver of crevasse propagation, and ultimately calving.

The study is concerned with a pressing topic in glaciology and climate science, and offers an innovative approach to the calving problem. A physically consistent combination of the FS model and LEFM under consideration of both surface and basal crevasses would constitute significant progress. The paper is furthermore clearly presented, well written, and nicely illustrated.

There are a few major points that I believe should be addressed. These are (i) a discussion of issues involving the complex rheology of ice, particularly that of combining a viscous flow model with an elastic fracture model (ii) the question of crevasse initiation; (iii) a more direct comparison with the observational crevasse data, and (iv) a more detailed analysis of the stress conditions that lead to crevasse propagation in LEFM. I will discuss these, together with a number of more minor comments, below.

Overall, I believe that this manuscript will — after some revisions — be a valuable contribution to the literature and a significant step toward adequately representing glacier calving in ice sheet and climate models.

Major Comments

1. The manuscript offers little discussion of the potential issues and limitations of its approach, and fails to put methods and results sufficiently into context with the existing literature. For one, LEFM is fundamentally somewhat at odds with the non-linear viscous-plastic rheology of glacier ice (e.g., Weiss 2004, Benn et al. 2007), a topic that merits at least some discussion. A particular issue is that of the different timescales of the viscous flow model and the LEFM crevasse propagation. In the paper, the LEFM routine is called every 20th time step of the FS model routine. Some justification or motivation for this particular coupling scheme should be given.

2. Previous studies have found that one of the main issues with applying LEFM is the requirement of sizable pre-existing fractures (Nath and Vaughan 2003, Krug et al. 2014). Here, on the other hand, crevasses can be initiated from infinitesimal cracks. Firstly, an explanation regarding the model features (in contrast to previous studies) that allow for such spontaneous initiations seems in order. Krug et al. (2014), for example, need to invoke a damage mechanics framework to "initiate" cracks.

Highlighting differences and similarities between the present model and that of Krug et al. (2014) —who also combine a Stokes model with LEFM — would be informative. Secondly, I must be misunderstanding something, because, if there's no initial threshold, it appears to me that this micro crack initiation would lead instantly to new crevasses wherever $K>K_c$? I understand that the authors are interested in the propagation, rather than initiation, of individual crevasses, but it seems that a physically plausible initiation process should be discussed in some form. Would a more complete model experiment initiate one crevasse at the location of maximum K, which then results in stress release in the vicinity, or similar?

3.  In the abstract, the authors state that they "find that FS/LEFM produces surface and bottom crevasses that match the distribution of [observed] crevasse depth and width". Although the text mentions some ball-park numbers for the different observed and modeled crevasse depths and widths, a more direct comparison is missing. I believe it would be helpful to add a figure that shows such a direct comparison, e.g, a width-vs-height scatterplot for observed and modeled crevasses, color coded by the different model experiments (or maybe some other, more suitable, illustration). It would also be interesting to see how observed and modeled crevasses compare as a function of distance from the grounding line.

4.  One of the key points of this study is that the HO and SSA approximations fail since, for these approximations, the stress intensity factor, K, does not surpass the fracture toughness beyond a limited crevasse depth. This is traced back to the hydrostatic assumption necessary for HO and SSA. It would be helpful to actually show a figure of how $K(t,x)$ evolves for the different models and different crevasse locations, and compare this to $K_c$. This may also help explain why "spontaneous" crevassing can occur in the model. It may even be worthwhile to show how the longitudinal stress, and ice and water pressure terms in (19) evolve. I was surprised to read that the crevasses in the HO and SSA simulations grew at all in the first place, and why are they then confined to a limit depth of 50 m? Maybe some further analysis of the evolution of K and a figure or two along the lines suggested above could help in the interpretation?

Minor Comments

**page 1**

l 24.   "iceberg calving *will* rise, which *will*" — replace with "is likely to rise, which would"? (This may not be as certain as the current version implies.)

**page 2**

l 7.   "each process" — which process? replace with "…the role of the different processes involved …" ?

l 8.   "where crevasses propagate"

l 11.   change to "… it is necessary to use a fracture theory, such as LEFM"? (There may be other ways; currently it sounds like this is the one and only)

**page 3**

Sec 2.2: It may be helpful to point out that (almost?) the same equations apply for the grounded and floating ice — only the boundary conditions differ.

l 20.   "values"

l 26.   add a reference for the 2D Full-Stokes model

**page 5**

l 3.   in previous work, (14) contains a N (pressure) factor. What happened to that?

l 9.   is $z_b(t)$ only unknown in FS, but not in HO an SSA?

l 15.   it is really the "ice-bedrock-ocean boundary"

l 17.   "migration of *the* grounding line"

l 28.   "avoids", or maybe better: "…term, invoked to avoid …"

**page 6**

l. 7   it's worth mentioning that K and the stresses depend on time and x (not just z)

l. 11   "equal"

l. 16   how is the width of the crevasse determined?

l. 18   a discussion of the coupling scheme between FS and LEFM model would be of interest here (see Major Point 1)

Sec 3.1. "FS model validation" —rather than evaluation (?) (see also l.26)

l. 27   "migration of *the* grounding line"

**page 7**

3.2. Model Setup: what's the duration of the integrations?

l. 1,2   "… *to* within X km"

l. 9     "In all  following experiments …"

l. 12    These are not really "micro" crevasses, but rather "infinitesimal" crevasses, no? Although the latter is of course more cumbersome to use.

l. 13    I found the choice of initial crevasse positions somewhat surprising. I presume the high density of crevasse locations near the grounding line was initially motivated to better resolve the potentially more variable crevasse dynamics in this region? It might be worth justifying this choice with half a sentence. Otherwise, I'd believe a linear or logarithmic spacing throughout the shelf may have made more sense.

l. 14    The "respectively" doesn't really work here. Maybe something like "In these experiments, the numbers 1—7 indicate crevasses initiated near the grounding line (at distances x = 0.5, …, 3.5km from the grounding line); the numbers 8, 9 indicate crevasses in the middle of the ice shelf (x = 18, 28 km); and numbers 10, 11 indicate crevasses near the ice front (x = 35, 36 km).

l. 21    "…positions for Experiments D and E are …"

**page 8**

Sec 4.2. — related to Major Comment 2: it would be nice to have some more information about the observed crevasses, i.e., how many, what's the spread, etc. I believe this could be summarized in a figure as suggested above.

l. 11    Can the authors elaborate on why the ice is "tens of meters below equilibrium" near the grounding line?

**page 9**

l. 5     (However, undercutting presumably occurs due to sustained ocean melt, i.e., it's not usually the case that undercutting occurs as a one-off sudden event, which then relaxes back to its normal state, but rather is maintained by a long-term temperature gradient in the water. So a fixed undercut profile might be more realistic.)

l. 16    equation (20) — not (19), same for l. 17

l. 20    "… grounding line *region* stop …"

Paragraph starting l.24: this is where a figure of the stresses depending on the model, experiment, and crevasse depth/shape would be insightful

l. 26    "crevasses"

**page 11**

l. 4     "crevasses"

**Figures**

**2.**     the blue lines are hard to see on my print version - choose green instead?
also differentiate the two red lines somehow more clearly?

**3.**     annotation: the β in the equation for bed drag should be $\alpha^2$ (or $\alpha^2 N$?)
caption: "… red line is the hydrostatic *bottom elevation* calculated …"

**5.**     color code the lines in (b) and the corresponding dots in (a), and then use
different symbols for Elmer/Ice and ISSM? Or at least put numbers onto the
profiles in (b). Also: in a dynamical systems context, the unstable equilibrium
state for the analytical solution (i.e. the negatively sloped part of the black curve)
is commonly given as a dashed line
caption: something seems to have gone wrong with the parentheses around the
references

**6.**     Why does the FS friction coefficient suddenly become large again at 0 (yellow
line overlaying the black box line)?

**7.**     There is this interesting behavior that the crack height for profiles 8–11 seems to
undergo a non-monotonic evolution, where it first jumps to a high value, then
decreases for a bit, before it recovers to a value similar to the original one. Can
the authors comment on that? Maybe a further analysis of the stresses as
suggested above may help with this interpretation as well?

Give it a title "Experiment A" and similarly for the panels of Figure **8** (to facilitate
an easy comparison with the text/ captions)

caption: "… b) Details"

**Additional References**

Weiss, J. "Subcritical crack propagation as a mechanism of crevasse formation and
iceberg calving." Journal of Glaciology 50.168 (2004): 109-115.

Nath, P. C., and D. G. Vaughan. "Subsurface crevasse formation in glaciers and ice
sheets." Journal of Geophysical Research: Solid Earth 108.B1 (2003).

---

## Referee Comment (RC2) · J. Bassis (Referee) · 20 Dec 2016

This study seeks to simulate surface and basal crevasse propagation using a variety of viscous models in an attempt to better understand the processes responsible for iceberg calving. The authors find that extra bending stresses near the grounding line that are resolved in full Stokes models are needed to accurately simulate the penetration depth of basal and surface crevasses. Moreover, the authors find that simulated crevasses are qualitatively similar to observed crevasses. Overall, I think this study is both interesting and novel and highly appropriate for publication in Cryosphere Discussion. I do, however, have two major points of concern along with a few more specific comments.

The first comment relates to the overarching goal of this study, which is presumably to better simulate iceberg calving. Here, I would like to see the authors make more detailed comparisons between observations and model predictions at the macro and local scale. For instance, the authors have strong conclusions about the role of different models in simulating crevasse penetration depths in Thwaites Glacier. But I had a hard time deciphering how well any of the models is able to reproduce the current ice tongue length of Thwaites. If the full Stokes model is able to accurately simulate the length of Thwaites Glacier tongue then this should be noted and celebrated. If the models have large discrepancies than, in the interest of honesty, this should also be acknowledged and discussed in light of (possible) missing processes.

Similarly, I would like to see the authors compare their simulated crevasse shapes with observed crevasse shapes (to the extent available). I realize that the radar probably doesn't fully resolve crevasse shapes, but one might be able to examine (at least to first order) penetration height and aspect ratios. Any such comparison would strengthen quantitative links between the model and observations, especially given the authors hints that such comparisons are positive.

My second comment focuses on the methodology and, more specifically, the description of the methodology. To start, it is not obvious to me why stresses computed using a viscous model can be used to drive an elastic fracture mechanics model. Here the authors could assist readers by summarizing the tenets of the theory that they are using to map viscous to elastic stresses or by computing elastic stresses and comparing them to viscous stresses. My uncertainty in the method goes a bit deeper here, both in the physical model posed and the numerical implementation. I will start by describing the questions I have about the physical model before outlining some more numerical concerns. None of these issues are fatal, but point towards a need for a more thorough description of the model assumptions and implementation.

Physical crevasse model description:

1. I think Equation (20) is wrong. If $\sigma_{xx}$ is the longitudinal stress then why is an additional cryostatic stress added? I'm also not sure that it is permissible in an elastic model to merely add a water pressure to the walls either without considering the effect on the stress field in the elastic body, but more on that later. Note also that Van der Veen does his computations using resistive stresses, not longitudinal or deviatoric stresses. The typo in Equation (20) makes it unclear to me that this translation has been done appropriately.

2. The assumption of elasticity and water (or air) filling crevasses tells us exactly the shape of the crevasse. Perhaps this is already taken into account, but I had a hard time figuring out the initial condition. Formally, we know that for an elastic material, the width of the crack is related to the pressure on the walls through an integral relationship (see, e.g., Equation 2.1 in Self-Similar Solutions for Elastohydrodynamic Cavity Flow by D. A. Spence and P. Sharp Proceedings of the Royal Society). Water pressure only enters into this equation through the expression for the water pressure on the crack walls. For a narrow aspect ratio crack (width to height), we can make the lubrication theory approximation, whence an equation for crack width (or rate of change for crack width) can be found. See for example, Lister (1990, Buoyancy-driven fluid fracture : similarity solutions for the horizontal and vertical propagation of fluid-filled cracks). Of course the water can also freeze on the walls, but I don't think the authors are dealing with thermodynamics.

3. Building on the previous point, the crevasse aspect ratio can be estimated by examining the ratio of pressure opening a crevasse to Young's modulus. For a 1 MPa stress opening a crevasse (an overestimate) and a typical 1-10 GPa Young's modulus I find that the aspect ratio of the crevasse is 0.0001-0.001. This translates into maximum crevasse widths (initially) that are of the order of $\sim$1 cm. For more typical stresses of the order of 100 kPa, I find initial widths of the order of a few mm. As a crude guess, one could take elliptical crevasses and evolve these, which is what I'm guessing the authors have done here, but this would seem to be incompatible with the 5 m resolution near the crevasses! More significantly, it shows that the crevasses the authors are considering (>200 m width) are \*\*incompatible\*\* with linear elastic fracture mechanics. Outside of the initial condition, I'm not convinced that LEFM is compatible with the fractures simulated. Of course, one might argue that new fractures are initiated at the tips of old fractures and the new fractures are elastic, but this needs to be made explicit.

4. These issues aside, even in the most simple LEFM implementations, we start with a starter crack. What size starter cracks is assumed?

Physical model implementation:

1. How are the crevasse implemented in ISSM? From the Figures I get the impression that crevasses are merely tracers and nodes are not actually removed when propagating crevasses. This needs to be better explained

2. If crevasse are implemented as tracers then they have no effect on the stress/strain rate field and this is probably fine for narrow crevasses, but is increasingly problematic for wide crevasses. We simulated crevasse evolution in Bassis and Ma (2015, 10.1016/j.epsl.2014.11.003) using an analytic calculation perturbation theory and found similar results to those shown, but found that the interaction between crevasses and the background strain rate was crucial to crevasse growth. This is particularly true for wide crevasses, which can result in accelerated flow into crevasses "healing" them. (We also found that large basal melt rates might erode basal crevasses, but this is more speculative.) More details are needed to assist readers to understand how the crevasses are initiated and evolved numerically. If this is a purely Lagrangian method, then this needs to be explained. If nodes are removed then this also should be explained. (There are thermodynamic and numerical issues associated with node removal, so these should be discussed if node removal is done.)

Technical comments:

Page 2, Line 10: The Nye zero stress model doesn't necessarily underestimate the

stress-concentration at the tip of crevasses. In fact, as Weertman and others have shown, the Nye zero stress model corresponds to the LEFM problem of closely spaced crevasses. Moreover, even for isolated crevasses, ice behaves like a viscous fluid over long time scales. Hence, it is unclear to me that crevasses should have a stress concentration that is equivalent to that in an elastic plate. In other words, it is unclear to me why the Nye zero stress model isn't the more physically accurate model over long time scales and/or for closely spaced crevasses.

Page 2, Line 25: Here the authors should comment on the appropriateness of a flowline model that neglects lateral drag. What width are the authors using for the flowline (or are they assuming the width is infinite)?

Page 3, Line 5: I don't understand the need for a damping term in the full Stokes equations. The buoyancy term on the bottom acts like a spring and, I thought, the solution, then includes the additional degree of freedom zb?

Equation 14: What is the physical reason for a linear sliding law? I thought prior research by Ian Joughin for Pine Island suggested a Weertman or plastic sliding law was most appropriate and would have (naively) thought that similar laws would be most appropriate for Thwaites as well? I apologize if I'm mistaken about this.

Page 8, line 20. So only crevasses wider than 200 m are considered, but these crevasses would already appear to violate the assumptions of the LEFM?

---

## Short Comment (SC1) · 4 Jan 2017

Nice study, just a minor point:

At page 6, line 20, the authors state: "Calving is assumed to occur when either the surface or the bottom crevasse reaches sea level (Benn et al., 2007)".

Calving when surface crevasses reach the waterline is justified by the resultant hydrofracturing. Why do the authors choose to prescribe calving when basal crevasses reach the waterline? I think it should be made clearer that this is a \*modification\* of the crevasse depth criterion proposed by Benn et al. (2007) and Nick et al. (2010).

---

## Author Comment (AC1) · 9 Feb 2017

Dear Editor and Reviewers,

Please find the attached supplement for our response to the review and changes on the manuscript.

Thanks.

Sincerely, Hongju Yu

Please also note the supplement to this comment:
http://www.the-cryosphere-discuss.net/tc-2016-249/tc-2016-249-AC1-supplement.pdf

---

## Author Response (AR1)

Dear Editor,

We are very thankful for the reviewers' comments. In response to these we have extensively revised the manuscript. We now explain in more detail how we combine LEFM and viscous flow, the process of crevasse initiation, the hypotheses we made and the limitations of our approach. We fixed minor typos, added Figure 7 and Figure 8, modified a number of figures to improve clarity, modified our description of the undercutting experiment, improve the comparison of modeled crevasses versus observed crevasses. These modifications improved the manuscript significantly but do not affect our conclusions. We hope you will find the paper acceptable for publication.

**1  Response to Reviewer Till Wagner**

Detailed below are our point-by-point responses to the comments of Reviewer Till Wagner. Reviewer's comments are printed in blue font followed by our responses in black.

*The authors develop a 2D ice-sheet model that includes the time-evolution of crevasses, by combining a Full Stokes (FS) model with an account of linear elastic fracture mechanics (LEFM). The results are compared to those obtained with higher-order (HO) and shallow-shelf approximations (SSA). The model is set up to represent a cross-section of Thwaites Glacier (TG), and the model output is compared to observed crevasses at TG. The results suggest that deviations from hydrostatic equilibrium are an important driver of crevasse propagation, and ultimately calving.*

*The study is concerned with a pressing topic in glaciology and climate science, and offers an innovative approach to the calving problem. A physically consistent combination of the FS model and LEFM under consideration of both surface and basal crevasses would constitute significant progress. The paper is furthermore clearly presented, well written, and nicely illustrated. There are a few major points that I believe should be addressed. These are (i) a discussion of issues involving the complex rheology of ice, particularly that of combining a viscous flow model with an elastic fracture model (ii) the question of crevasse initiation; (iii) a more direct comparison with the observational crevasse data, and (iv) a more detailed analysis of the stress conditions that lead to crevasse propagation in LEFM. I will discuss these, together with a number of more minor comments, below.*

*Overall, I believe that this manuscript will - after some revisions - be a valuable contribution to the literature and a significant step toward adequately representing glacier calving in ice sheet and climate models.*

*1. The manuscript offers little discussion of the potential issues and limitations of its approach, and fails to put methods and results sufficiently into context with the existing literature. For one, LEFM is fundamentally somewhat at odds with the non-linear viscous-plastic rheology of glacier ice (e.g., Weiss (2004); Benn et al. (2007), a topic that merits at least some discussion. A particular issue is that of the different timescales of the viscous flow model and the LEFM crevasse propagation. In the paper, the LEFM routine is called every 20th time step of the FS model routine. Some justification or motivation for this particular coupling scheme should be given.*

Yes, the LEFM and viscous theories act on different time scales: the LEFM is essentially instantaneous (or the speed of sound) while the viscous flow takes place on the time scale of days to months. It is also correct that the LEFM cannot explain some aspects of crevasse propagation. Weiss (2004) argued that LEFM cannot explain the initiation of crevasse from mm to m scale and proposed a subcritical crack propagation method. Here, we do not need this method because our focus is the propagation of crevasses, not their initiation. However, this means that our LEFM model requires a minimum crevasse depth. Here, we assume that the crevasses propagate from infinitesimal crevasses that are always present in glacier ice and we assume they can propagate if their required depth is less than 1 m. According to LEFM, the width of crevasse should be in the order of $\sim$1 cm. It is computational too demanding to have mesh resolution at this scale. We therefore assume that the crevasse grow to 20 m instantly after its opening. In our results, the crevasse grow to 60–70 m after only 0.01 yr due to viscous deformation. Beyond the crevasse initiation process, the LEFM theory has been used successfully to model crevasse propagation (van der Veen, 1998a,b), rift propagation (Larour et al., 2004a,b), and calving (Krug et al., 2014) in prior studies.

In terms of time step, the LEFM model is called every 0.01 yr because the time scale of calving events is on the order of days to weeks (James et al., 2014; Murray et al., 2015). We investigated the impact of the time step by running the same experiment but calling the LEFM model every 0.005 yr for 0.15 yr (to save computational time and as the crevasse depth is stable after 0.15 yr). As shown in Fig. 1 below, the differences in crevasse depth are less than 20 m in all the experiments. The time step chosen for the LEFM therefore has no significant impact on the results. The limitations of this combination is now discussed in the manuscript at page 7 line 16–25 and the choose of time step is discussed at page 7, line 8–10. The figure is added to the supplementary material as Fig. S1.

*2. Previous studies have found that one of the main issues with applying LEFM is the requirement of sizable pre-existing fractures (Nath and Vaughan 2003, Krug et al. 2014). Here, on the other hand, crevasses can be initiated from infinitesimal cracks. Firstly, an explanation regarding the model features (in contrast to previous studies) that allow for*

*such spontaneous initiations seems in order. Krug et al. (2014), for example, need to invoke a damage mechanics framework to "initiate" cracks. Highlighting differences and similarities between the present model and that of Krug et al. (2014) who also combine a Stokes model with LEFM - would be informative. Secondly, I must be misunderstanding something, because, if there's no initial threshold, it appears to me that this micro crack initiation would lead instantly to new crevasses wherever K>Kc? I understand that the authors are interested in the propagation, rather than initiation, of individual crevasses, but it seems that a physically plausible initiation process should be discussed in some form. Would a more complete model experiment initiate one crevasse at the location of maximum K, which then results in stress release in the vicinity, or similar?*

Agreed. As mentioned above, with LEFM, the criterion $K > K_c$ is never satisfied when the crevasse depth is small (cm scale). A minimum depth is therefore required. Here, we start from an infinitesimal crevasse (always present) and assume that the crevasse can propagate if its required depth is smaller than 1 m. The reason we place the initial crevasse at different locations is to see how the propagation varies as a function of distance to the grounding line. This is now discussed in the manuscript at page 7, line 16–20.

We do not attempt to predict the initial crevasse location, for instance based on the initial K value. Radar echograms also indicate that crevasses are densely distributed on the ice shelf of Thwaites.

In Krug et al. (2014), damage mechanics is used to determine where a crevasse grows, but they do not discuss the crevasse propagation process. Their crevasses will only propagate if they penetrate the whole ice thickness to create a calving event. Here, we focus entirely on the propagation of crevasses. This is now pointed out in the manuscript at page 2, line 17–19.

*3. In the abstract, the authors state that they "find that FS/LEFM produces surface and bottom crevasses that match the distribution of [observed] crevasse depth and width". Although the text mentions some ball-park numbers for the different observed and modeled crevasse depths and widths, a more direct comparison is missing. I believe it would be helpful to add a figure that shows such a direct comparison, e.g, a width-vs-height scatterplot for observed and modeled crevasses, color coded by the different model experiments (or maybe some other, more suitable, illustration). It would also be interesting to see how observed and modeled crevasses compare as a function of distance from the grounding line.*

A width-vs-height scatter plot has been added in the manuscript (Fig. 7). The depth of the observed and modeled crevasses match well. Some observed crevasses are wider than our modeled crevasses because: 1) at the end of our experiments, the crevasse depth is stable but the width is still increasing, hence our model stops too soon; and 2) ocean forcing is

not included in the model, which could affect crevasse growth. This discussion of the figure is added in the manuscript at page 10, line 23–25.

*4. One of the key points of this study is that the HO and SSA approximations fail since, for these approximations, the stress intensity factor, K, does not surpass the fracture toughness beyond a limited crevasse depth. This is traced back to the hydrostatic assumption necessary for HO and SSA. It would be helpful to actually show a figure of how K(t,x) evolves for the different models and different crevasse locations, and compare this to Kc. This may also help explain why "spontaneous" crevassing can occur in the model. It may even be worthwhile to show how the longitudinal stress, and ice and water pressure terms in (19) evolve. I was surprised to read that the crevasses in the HO and SSA simulations grew at all in the first place, and why are they then confined to a limit depth of 50 m? Maybe some further analysis of the evolution of K and a figure or two along the lines suggested above could help in the interpretation?*

Agreed. We have now added a figure in the manuscript (Fig. 8) to show the relationship between K and crevasse depth with different models, different initial crevasse positions and different time. Crevasses never propagate in HO and SSA if the initial crevasse position is >2000 m downstream of grounding line. In these cases, the temporal changes in K are small because the geometry and thus the velocity and stress field do not change significantly. When the initial crevasse positions are within 2000 m of the grounding line, the crevasses propagate. Then, K decreases, the crevasses stop propagating and close up because of viscous flow. We think 50 m is only a limit for our specific flowline. It may vary for a different glacier with a different stress regime. This discussion of the figure is added in the manuscript at page 9, line 22–25.

*Minor Comments*

*page 1*

*l 24. "iceberg calving will rise, which will" replace with "is likely to rise, which would"? (This may not be as certain as the current version implies.)*

Done. The manuscript is modified at page 1, line 24.

*page 2*

*l 7. "each process" which process? replace with "...the role of the different processes involved ..." ?*

Done. The manuscript is modified at page 2, line 6–8.

*l 8. "where crevasses propagate"*

Done. The manuscript is modified at page 2, line 9.

*l 11. change to "... it is necessary to use a fracture theory, such as LEFM"? (There may be other ways; currently it sounds like this is the one and only)*

Done. The manuscript is modified at page 2, line 14.

*page 3*

*Sec 2.2: It may be helpful to point out that (almost?) the same equations apply for the grounded and floating ice - only the boundary conditions differ.*

Done. The manuscript is modified at page 3, line 19–20.

*l 20. "values"*

Done. The manuscript is modified at page 3, line 27.

*l 26. add a reference for the 2D Full-Stokes model*

Done. The manuscript is modified at page 4, line 6.

*page 5*

*l 3. in previous work, (14) contains a N (pressure) factor. What happened to that?*

Many sliding laws have been proposed in the past. Studies have shown that a Weertman sliding law (basal drag is non-linearly related to ice velocity) or a sliding law using a N factor (effective pressure at the bed) may be more appropriate than a linear sliding law in simulating glacier dynamics (Lliboutry, 1987; Cuffey and Paterson, 2010). Here, however, the time step and total simulation time are short. The grounding line does not migrate and the changes in ice thickness are small, so the influence of the particular sliding law we use is limited. After the inversion, the modeled velocity matches the observed velocity well and the background velocity field does not change significantly during the simulations. We therefore choose to use the linear sliding law for simplicity. This is now discussed in manuscript at page 5, line 13–17.

*l 9. is $z_b(t)$ only unknown in FS, but not in HO an SSA?*

In FS, if we do not use a dampening term, the vertical velocity becomes unrealistically high ($\sim 10^5$ m/yr) and destabilizes the system. In HO and SSA, $z_b(t)$ is not an unknown because we solve for ice thickness and use hydrostatic equilibrium to calculate the ice surface and bottom elevation. In these cases, the vertical velocity is decoupled from the system. Hence no dampening term is needed for HO and SSA. This is now discussed in the manuscript in page 5, line 26–27.

*l 15. it is really the "ice-bedrock-ocean boundary"*

Done. The manuscript is modified at page 5, line 29.

*l 17. "migration of the grounding line"*

Done. The manuscript is modified at page 6, line 2.

*l 28. "avoids", or maybe better: "...term, invoked to avoid ..."*

Done. The manuscript is modified at page 6, line 13.

*page 6*

*l.7 it's worth mentioning that K and the stresses depend on time and x (not just z)*

Done. The manuscript is modified at page 6, line 19.

*l. 11 "equal"*

Done. The manuscript is modified at page 6, line 26.

*l.16 how is the width of the crevasse determined?*

The crevasse is assumed to grow to 20 m wide once it opens. Then it becomes wider due to the viscous flow. This is now mentioned in the manuscript at page 7, line 4–5.

*l.18 a discussion of the coupling scheme between FS and LEFM model would be of interest here (see Major Point 1)*

We added a paragraph in section 2.5 (page 7, line 16–25) discussing the limitation of the combination of LEFM and viscous flow, namely the requirement of a minimum crevasse

depth and the difference in the scale between a crevasse modeled using LEFM and using viscous flow.

*Sec 3.1. "FS model validation" rather than evaluation (?) (see also l.26)*

Done. The manuscript is modified at page 7, line 27 and line 30.

*l. 27 "migration of the grounding line"*

Done. The manuscript is modified at page 7, line 31.

*page 7*

*3.2. Model Setup: what's the duration of the integrations?*

The duration is 0.3 yr. This is now added in the manuscript at page 8, line 11–12.

*l. 1,2 "... to within X km"*

Done. The manuscript is modified at page 8, line 3 and line 4.

*l. 9 "In all  following experiments ..."*

Done. The manuscript is modified at page 8, line 12.

*l.12 These are not really "micro" crevasses, but rather "infinitesimal" crevasses, no? Although the latter is of course more cumbersome to use.*

Done. The term "micro crevasses" is changed to "infinitesimal crevasses" throughout the paper.

*l.13 I found the choice of initial crevasse positions somewhat surprising. I presume the high density of crevasse locations near the grounding line was initially motivated to better resolve the potentially more variable crevasse dynamics in this region? It might be worth justifying this choice with half a sentence. Otherwise, I'd believe a linear or logarithmic spacing throughout the shelf may have made more sense.*

This is correct. The stress field is complex and exhibits large variations near the grounding line and this is why we choose the initial crevasse positions more densely in the grounding line region. This is now pointed out in the manuscript at page 8, line 18–20.

*l.14 The "respectively" doesn't really work here. Maybe something like "In these experiments, the numbers 1-7 indicate crevasses initiated near the grounding line (at distances x = 0.5, ..., 3.5km from the grounding line); the numbers 8, 9 indicate crevasses in the middle of the ice shelf (x = 18, 28 km); and numbers 10, 11 indicate crevasses near the ice front (x = 35, 36 km).*

Done. The manuscript is modified at page 8, line 16–18.

Done. The manuscript is modified at page 8, line 26.

*Sec 4.2. related to Major Comment 2: it would be nice to have some more information about the observed crevasses, i.e., how many, what's the spread, etc. I believe this could be summarized in a figure as suggested above.*

The figure is added to the manuscript as Fig. 7.

*l.11 Can the authors elaborate on why the ice is "tens of meters below equilibrium" near the grounding line?*

This displacement below hydrostatic equilibrium is caused by a bending moment of the ice that is resulted from the abrupt change in basal conditions at the grounding line. This is now pointed out in the manuscript at page 9, line 14–15.

*l. 5 (However, undercutting presumably occurs due to sustained ocean melt, i.e., it's not usually the case that undercutting occurs as a one-off sudden event, which then relaxes back to its normal state, but rather is maintained by a long-term temperature gradient in the water. So a fixed undercut profile might be more realistic.)*

We run the undercutting experiments again by adding a 3000 m/yr basal melting in the undercutting region to achieve an almost fixed undercutting shape during the simulations. The results are shown in Fig.2 below. The results is similar to our previous in terms of crevasse depth and whether calving happens. If the high melt cannot sustain, the undercutting front diminishes in $\sim$0.1 yr. If the high melt can sustain, which is dependent on the seasonal variability of the ocean conditions, then the undercutting shape can sustain for a longer time. This is now discussed in the manuscript at page 12, line 1–3.

Done. The manuscript is modified at page 10, line 27 and line 28.

*l. 20 "... grounding line region stop ..."*

Done. The manuscript is modified at page 10, line 31.

*Paragraph starting l.24: this is where a figure of the stresses depending on the model, experiment, and crevasse depth/shape would be insightful*

The figure is added to the manuscript as Fig. 8.

*l. 26 "crevasses"*

Done. The manuscript is modified at page 11, line 5.

*l. 4 "crevasses"*

Done. The manuscript is modified at page 12, line 21.

*Figures*

*2. the blue lines are hard to see on my print version – choose green instead? also differentiate the two red lines somehow more clearly?*

Done. Figure 2 is modified at page 18. We changed the blue lines to yellow lines and the lower red lines to green lines. The grounding line positions are now marked using cyan dots.

*3. annotation: the $\beta$ in the equation for bed drag should be $\alpha^2$ (or $\alpha^2 N$?) caption: "... red line is the hydrostatic bottom elevation calculated ..."*

Done. Figure 3 is modified at page 19.

*5. color code the lines in (b) and the corresponding dots in (a), and then use different symbols for Elmer/Ice and ISSM? Or at least put numbers onto the profiles in (b). Also:*

*in a dynamical systems context, the unstable equilibrium state for the analytical solution (i.e. the negatively sloped part of the black curve) is commonly given as a dashed line*

*caption: something seems to have gone wrong with the parentheses around the references*

Done. Figure 5 is modified at page 20.

*6. Why does the FS friction coefficient suddenly become large again at 0 (yellow line overlaying the black box line)?*

One node after the grounding line was mistakenly plotted with a default value, which was never used since it is floating. Figure 6 is now fixed at page 20. Thank you for pointing this out.

*7. There is this interesting behavior that the crack height for profiles 8-11 seems to undergo a non-monotonic evolution, where it first jumps to a high value, then decreases for a bit, before it recovers to a value similar to the original one. Can the authors comment on that? Maybe a further analysis of the stresses as suggested above may help with this interpretation as well?*

When the crevasses propagate, K will decrease. In profiles 1–4, the crevasses can continue to propagate after this decrease so they show a monotonic evolution through the simulation. In profile 5–11, however, the crevasses cannot continue to propagate after this decrease. Then, when the crevasses become shallower from viscous deformation, K increases until the crevasse can grow again. This is now mentioned in the manuscript at page 10, line 16–19.

*Give it a title "Experiment A" and similarly for the panels of Figure 8 (to facilitate an easy comparison with the text/ captions)*

*caption: "... b) Details"*

Done. Figure 9 (previous figure 7) is modified at page 22.

**2   Response to Reviewer Jeremy Bassis**

Detailed below are our point-by-point responses to the comments of Reviewer Jeremy Bassis. Reviewer comments are printed in blue font followed by our responses in black.

*This study seeks to simulate surface and basal crevasse propagation using a variety of viscous models in an attempt to better understand the processes responsible for iceberg calving. The authors find that extra bending stresses near the grounding line that are resolved in full Stokes models are needed to accurately simulate the penetration depth of basal and surface crevasses. Moreover, the authors find that simulated crevasses are qualitatively similar to observed crevasses. Overall, I think this study is both interesting and novel and highly appropriate for publication in Cryosphere Discussion. I do, however, have two major points of concern along with a few more specific comments.*

*The first comment relates to the overarching goal of this study, which is presumably to better simulate iceberg calving. Here, I would like to see the authors make more detailed comparisons between observations and model predictions at the macro and local scale. For instance, the authors have strong conclusions about the role of different models in simulating crevasse penetration depths in Thwaites Glacier. But I had a hard time deciphering how well any of the models is able to reproduce the current ice tongue length of Thwaites. If the full Stokes model is able to accurately simulate the length of Thwaites Glacier tongue then this should be noted and celebrated. If the models have large discrepancies than, in the interest of honesty, this should also be acknowledged and discussed in light of (possible) missing processes.*

We do not attempt to predict the length of the ice tongue with any of the models. First, we do not have a process to determine where the crevasse initiates. We choose its position arbitrarily to study how crevasses propagate at different locations. Second, the length of the ice tongue depends on many other factors, such as ocean currents (which slowly rotate the tongue to the west), the ice melange, buttressing from the sides, etc. We could predict the length of the ice tongue by using damage mechanics or subcritical crevasse propagation (Krug et al., 2014; Weiss, 2004), but this is beyond the scope of this study. Elements of this discussion are now added at page 12, line 14–17 as future work. Thank you for the comment.

*Similarly, I would like to see the authors compare their simulated crevasse shapes with observed crevasse shapes (to the extent available). I realize that the radar probably doesn't fully resolve crevasse shapes, but one might be able to examine (at least to first order) penetration height and aspect ratios. Any such comparison would strengthen quantitative links between the model and observations, especially given the authors hints that such comparisons are positive.*

A width-vs-height scatterplot for observed and modeled crevasses is added in the manuscript as Fig. 7. The depth of observed and modeled crevasses match well. Some observed crevasses are wider than our modeled crevasse width because: 1) at the end of our experiments, the crevasse depth is stable but the width is still increasing, hence our model stops

too soon; and 2) ocean forcing is not included in the model, which could affect crevasse growth. This discussion of the figure is added in the manuscript at page 10, line 23–25.

*My second comment focuses on the methodology and, more specifically, the description of the methodology. To start, it is not obvious to me why stresses computed using a viscous model can be used to drive an elastic fracture mechanics model. Here the authors could assist readers by summarizing the tenets of the theory that they are using to map viscous to elastic stresses or by computing elastic stresses and comparing them to viscous stresses. My uncertainty in the method goes a bit deeper here, both in the physical model posed and the numerical implementation. I will start by describing the questions I have about the physical model before outlining some more numerical concerns. None of these issues are fatal, but point towards a need for a more thorough description of the model assumptions and implementation.*

*Physical crevasse model description:*

*1. I think Equation (20) is wrong. If $\sigma_{xx}$ is the longitudinal stress then why is an additional cryostatic stress added? I'm also not sure that it is permissible in an elastic model to merely add a water pressure to the walls either without considering the effect on the stress field in the elastic body, but more on that later. Note also that Van der Veen does his computations using resistive stresses, not longitudinal or deviatoric stresses. The typo in Equation (20) makes it unclear to me that this translation has been done appropriately.*

This was a typo. The stress we used is the deviatoric stress: $\sigma'_{xx} = \sigma_{xx} + p$. The resistive stress used in van der Veen (1998b) is described in van der Veen and Whillans (1989): $R_{xx} = \sigma_{xx} + L$, where $R_{xx}$ is the resistive stress and $L = \rho g h$ is the lithostatic pressure. In the simulation, the calculated pressure $p$ is only deviated from the lithostatic pressure by the order of 1%. Therefore, we do not think this difference has an impact on our results. The equation is now fixed at page 6, line 23.

*2. The assumption of elasticity and water (or air) filling crevasses tells us exactly the shape of the crevasse. Perhaps this is already taken into account, but I had a hard time figuring out the initial condition. Formally, we know that for an elastic material, the width of the crack is related to the pressure on the walls through an integral relationship (see, e.g., Equation 2.1 in Self-Similar Solutions for Elastohydrodynamic Cavity Flow by D. A. Spence and P. Sharp Proceedings of the Royal Society). Water pressure only enters into this equation through the expression for the water pressure on the crack walls. For a narrow aspect ratio crack (width to height), we can make the lubrication theory approximation, whence an equation for crack width (or rate of change for crack width) can be found. See for example, Lister (1990, Buoyancy-driven fluid fracture : similarity solutions for the horizontal and vertical propagation of fluid-filled cracks). Of course the water can also freeze on the walls, but I don't think the authors are dealing with thermodynamics.*

*3. Building on the previous point, the crevasse aspect ratio can be estimated by examining the ratio of pressure opening a crevasse to Young's modulus. For a 1 MPa stress opening a crevasse (an overestimate) and a typical 1-10 GPa Young's modulus I find that the aspect ratio of the crevasse is 0.0001-0.001. This translates into maximum crevasse widths (initially) that are of the order of ~1 cm. For more typical stresses of the order of 100 kPa, I find initial widths of the order of a few mm. As a crude guess, one could take elliptical crevasses and evolve these, which is what I'm guessing the authors have done here, but this would seem to be incompatible with the 5 m resolution near the crevasses! More significantly, it shows that the crevasses the authors are considering (>200 m width) are \*\*incompatible\*\* with linear elastic fracture mechanics. Outside of the initial condition, I'm not convinced that LEFM is compatible with the fractures simulated. Of course, one might argue that new fractures are initiated at the tips of old fractures and the new fractures are elastic, but this needs to be made explicit.*

Since both points 2 and 3 concern the crevasse width, we combined our response. The width of the crevasses estimated from LEFM is ~1 cm. However, once the crevasses are formed elastically, their shape is controlled by viscous flow, which grows the crevasses wider. Numerically, we cannot afford to run simulations with a mesh resolution of 1 cm. We therefore assume that the crevasses grow to 20 m wide after initiation. In the end, the crevasses grow from 20 to 60-70 m wide in 0.01 yr, so we think that it is reasonable to expect that the initial crevasse to grow to 20 m wide rapidly. When the LEFM method is employed again, we do not consider pre-existing crevasses because their width violate the assumption of LEFM. The pre-existing crevasses are only considered to be a feature of the ice shelf and affect stress field computed from the viscous model. The LEFM method is applied to an infinitesimal crevasse at the apex of the pre-existing crevasses. This new crevasse is opened to 20 m instantly and then it merges into the pre-existing crevasses due to viscous flow. This is now discussed in the manuscript from page 7, line 10–14 and line 20–25.

*4. These issues aside, even in the most simple LEFM implementations, we start with a starter crack. What size starter cracks is assumed?*

When using the LEFM, as mentioned above, the criterion K > Kc is never satisfied when the crevasse depth is small (cm scale) and a minimum required crevasse depth is needed. Here, we start from an infinitesimal crevasse (always present) and assume that the crevasse can propagate if its required depth is smaller than 1 m. This is now discussed in the manuscript at page 7, line 16–20.

*Physical model implementation:*

*1. How are the crevasse implemented in ISSM? From the Figures I get the impression*

*that crevasses are merely tracers and nodes are not actually removed when propagating crevasses. This needs to be better explained*

Once the crevasse depth is computed using LEFM, we change the surface and bottom elevation of the flowline to represent the crevasse propagation. Numerically, this is done by migrating the nodes vertically. In the entire simulation, none of the node is removed from the mesh. This is now mentioned in the manuscript at page 7, line 6–7.

*2. If crevasse are implemented as tracers then they have no effect on the stress/strain rate field and this is probably fine for narrow crevasses, but is increasingly problematic for wide crevasses. We simulated crevasse evolution in Bassis and Ma (2015, 10.1016/j.epsl.2014.11.003) using an analytic calculation perturbation theory and found similar results to those shown, but found that the interaction between crevasses and the background strain rate was crucial to crevasse growth. This is particularly true for wide crevasses, which can result in accelerated flow into crevasses "healing" them. (We also found that large basal melt rates might erode basal crevasses, but this is more speculative.) More details are needed to assist readers to understand how the crevasses are initiated and evolved numerically. If this is a purely Lagrangian method, then this needs to be explained. If nodes are removed then this also should be explained. (There are thermodynamic and numerical issues associated with node removal, so these should be discussed if node removal is done.)*

With the modified geometry after the crevasse shape is computed, the initial conditions of the viscous model are modified and thus the stress field is modified. The "healing" effect is also observed in our experiments as the crevasses become shallower and wider. This mentioned in the manuscript at page 7, line 5–8.

*Technical comments:*

*Page 2, Line 10: The Nye zero stress model doesn't necessarily underestimate the stress-concentration at the tip of crevasses. In fact, as Weertman and others have shown, the Nye zero stress model corresponds to the LEFM problem of closely spaced crevasses. Moreover, even for isolated crevasses, ice behaves like a viscous fluid over long time scales. Hence, it is unclear to me that crevasses should have a stress concentration that is equivalent to that in an elastic plate. In other words, it is unclear to me why the Nye zero stress model isn't the more physically accurate model over long time scales and/or for closely spaced crevasses.*

We agree that the Nye zero stress model is similar to LEFM with closely spaced crevasses. However, it has been shown that closely spaced crevasses will produce shallower crevasses than isolated crevasses (van der Veen, 1998b). Over a long time scale, there will not be any stress concentration in ice, but on a short time scale, especially when the crevasse

propagates, stress concentration exists at the tip of crevasses. This is discussed in the manuscript at page 2, line 11-13.

*Page 2, Line 25: Here the authors should comment on the appropriateness of a flowline model that neglects lateral drag. What width are the authors using for the flowline (or are they assuming the width is infinite)?*

The lateral drag is parameterized following Gagliardini et al. (2010). The width of the glacier is assumed to be 130 km from observations. This is described in the manuscript from page 4, line 27 to page 5, line 1.

*Page 3, Line 5: I don't understand the need for a damping term in the full Stokes equations. The buoyancy term on the bottom acts like a spring and, I thought, the solution, then includes the additional degree of freedom $z_b$?*

The dampening term is used to stabilize the model. Without this dampening term, the vertical velocity computed from FS model becomes unrealistically high ($\sim 10^5$ m/yr) and destabilizes the system. In HO and SSA, $z_b(t)$ is not an unknown because we solve for ice thickness and use hydrostatic equilibrium to calculate the ice surface and bottom elevation. In these cases, the vertical velocity is decoupled from the system. Hence the dampening term is not needed for HO and SSA. The manuscript is modified at page 5, line 26-27.

*Equation 14: What is the physical reason for a linear sliding law? I thought prior research by Ian Joughin for Pine Island suggested a Weertman or plastic sliding law was most appropriate and would have (naively) thought that similar laws would be most appropriate for Thwaites as well? I apologize if I'm mistaken about this.*

Some prior studies have shown that a Weertman sliding law (basal drag is non-linearly related to the ice velocity) or a sliding law considering a N factor (effective pressure at the bed) may be more appropriate than a linear sliding law in simulating glacier dynamics (Lliboutry, 1987; Cuffey and Paterson, 2010). Here, however, the time step and total simulation time are short. The grounding line does not migrate and the changes in ice thickness are small, so the influence of the particular sliding law we use is limited. After the inversion, the modeled velocity matches the observed velocity well and the background velocity field does not change significantly during the simulations. Therefore, we choose to use the linear sliding law for simplicity. This is now discussed in manuscript at page 5, line 1317.

*Page 8, line 20. So only crevasses wider than 200 m are considered, but these crevasses would already appear to violate the assumptions of the LEFM?*

We only consider crevasses wider than 200 m because the depth and width of observed crevasses are not estimated so precisely from the radar echograms. For smaller crevasses, the estimation error is high and the uncertainties in shape is large. In our model, LEFM is only responsible for the propagation of crevasse; the change of crevasse width is controlled by the viscous deformation of ice. The manuscript is modified at page 9, line 6–7.

**3    Response to Short Comment from Joe Todd**

*Nice study, just a minor point:*

*At page 6, line 20, the authors state: "Calving is assumed to occur when either the surface or the bottom crevasse reaches sea level (Benn et al., 2007)". Calving when surface crevasses reach the waterline is justified by the resultant hydrofracturing. Why do the authors choose to prescribe calving when basal crevasses reach the waterline? I think it should be made clearer that this is a \*modification\* of the crevasse depth criterion proposed by Benn et al. (2007) and Nick et al. (2010).*

This is a good point. The calving criterion of bottom crevasses has been changed. Now, calving only occurs when the bottom crevasse meets the surface crevasse or when the surface crevasse reaches sea level. This does not affect our results. In our experiments, every calving case except Exp. D2 and Exp. D3 has a bottom crevasse that penetrate not only the waterline but also the ice surface. For Exp. D2 and Exp. D3, the experiments were rerun with this new calving criterion. Calving still occurs, only few time steps later. We modified the description of the calving criterion in the manuscript at page 7, line 14–15. Figure 10 (previously figure 8) is also modified using the results obtained with this calving criterion.

[Figure]

Figure 1: a) Crevasse propagation with different time steps. The blue lines are the shape of crevasses after 0.15 yr with the LEFM routine called every 0.01 yr. The red lines are the shape of the crevasses after 0.15 yr with the LEFM routine called every 0.005 yr. b) Zoom in to the grounding line region.

[Figure]

Figure 2: a) Crevasse propagation with undercutting. The blue lines are the shape of crevasses with no additional melt (Exp. E1–7). The red lines are the shape of the crevasses with 3000 m/yr additional melt the in undercutting region.

[revised manuscript text omitted]

---

## Referee Report (RR1)

Overall, I think the authors have done a commendable job of addressing reviewer comments. In addition to a few technical corrections, I still have two questions/comments, but these may reflect my own misunderstanding of the methods used by the authors and I leave it at the discretion of the editors and authors whether these comments need to be addressed prior to publication. I also strongly suggest that the authors provide inputs to their model as supplementary data tables or figures. I think most of the information is included as figures, but I don't think the authors provide the mass balance field they use and this should be included as a figure or as a supplementary table. Personally, I would advocate that the authors include all inputs (basal friction, mass balance, bed geometry, etc) as a data table supplementary information to allow others to more easily reproduce the authors results. My major comments refer to previous comments and are labeled as such.

**Response to response to comment 1:** I'm still slightly concerned about the distinction between deviatoric and resistive stresses which is explored in the response to my comment from the previous review. I've pondered this and I think I'm missing something here. As I understand it, the authors substitute deviatoric stresses into Van der Veen's resistive stress formulation to calculate stress intensities. This is justified based on the fact that the pressure only deviates by a small amount from the hydrostatic pressure (cryostatic, lithostatic, what ever). The definition of resistive stresses given by the authors in the reply appears to diverge from Van der Veen's. Van der Veen defines $R_{xx} = \sigma_{xx} + \rho g\,(h_s - z)$ not $R_{xx} = \sigma_{xx} + \rho g H$, where $h_s$ denotes the surface elevation and $h$ is the ice thickness. I assume this is a typo in the authors reply (?). Nonetheless, my understanding is that the authors argue that deviatoric stress $\sigma'_{xx} = \sigma_{xx} + p$ is (nearly) equivalent to resistive stresses when pressure $p$ is (nearly) hydrostatic, i.e., $p = \rho g(h_s - z)$ and the authors argue that pressure is in fact within a few percent of hydrostatic. However, in the shelfy-stream approximation, the vertical force balance condition requires that $p = \rho g\,(h - z) - \left(\sigma'_{xx} + \sigma'_{yy}\right)$. At least in the shallow approximation, the pressure is reduced by horizontal stretching. In the interior of the ice sheet, where the ice is frozen to its bed, longitudinal stretching is small and pressure is approximately hydrostatic. A consequence is that $2\sigma'_{xx} = R_{xx}$ (again in the shallow shelf limit). This argument only holds for the shallow models and doesn't hold for the full Stokes calculation, but I would expect qualitatively similar results in which deviatoric stresses act to reduce the pressure. This leaves me confused as to what the authors did to calculate crevasse depths and why the pressure in their model is (nearly) hydrostatic. I assume the authors did the right thing, but it is hard to decipher what that thing is.

**Response to response to comment 2:** This is probably reasonable, but again I'm still slightly confused. If crevasses have initial width $w_0$, then the rate at which they widen initially will be proportional to the extensional strain rate, say $dw/dt = E_{xx}w$, where $E_{xx}$ is a measure of the extensional strain rate opening the crevasse. This results in an exponential widening rate early on in

the evolution of the crevasse. This suggests that if crevasses are initially 20 m wide and the width triples to 60 m in time interval $\Delta t$, then a 1 cm wide (my calculations suggest more like 4-6 mm) should widen to 3 cm in $\Delta t$ just based on the kinematics of the flow field. This would suggest that the width of crevasses later on in the simulation may be a function of the initial width of the crevasses and better (or worse) agreement with observations could be obtained merely by adjusting the initial width of the crevasse. Again, I assume the authors have done grid sensitivity experiments, but a statement or two (or a plot) to point out that the width of crevasses throughout the simulation is only weakly dependent on numerical resolution would be useful to fortify this in the audiences mind.

**Technical corrections**
I still have questions about the role of damping in simulating the stress field (especially as it translates to crevasse depths). The damping term can create an additional stress that acts to open crevasse because the ice shelf is never exactly in hydrostatic equilibrium with the ocean. However, this may be a higher-order numerical question.

Page 7, line 10: The time scale of major calving events from ice shelves is years-to-decades, not days to weeks. The days to weeks time frame is appropriate for grounded Greenland glaciers.

---

## Author Response (AR2)

Dear Editor,

We are very thankful for the reviewers' comments. We have revised the manuscript accordingly. We added a few sentences, fixed typos and clarified a few statements. We improved the figures to better illustrate our results. We hope you will find the revised paper acceptable for publication.

**1 Response to Reviewer Till Wagner**

Detailed below are our point-by-point responses to the comments from reviewer Till Wagner. Reviewer's comments are in blue, followed by our responses in black.

*I commend the reviewers for comprehensively addressing my comments. I believe the manuscript to be much improved and regard it worthy of publication, subject to some minor revisions (which I will detail below). Overall, I still found a number of typos and grammatical errors (some of which I'll point out below), and I urge the authors to address these issues in the next round of revisions.*

*page 1*

*l.5 "match the distribution" appears somewhat strong, since the observed crevasse dimensions are significantly more spread out than the modeled ones. I suggest "are consistent with the distribution". Furthermore, the authors discuss the mismatch of the observed and modeled crevasse widths, but there is no mention of the modeled surface crevasses being restricted to much smaller depths than the observed distribution. An acknowledgment of this mismatch (and speculative explanation?) may be in order.*

Agreed. We changed "match the distribution" to "are consistent with the distribution" in the manuscript at page 1, line 5. We agree the observed surface crevasses are deeper, but a number of these correspond to rift, i.e. ice shelf cut all the way to sea level, hence nearly equivalent to calving events. Bottom crevasses are similar in height but narrower. We note however that we only let the crevasses evolve for 0.3 yr whereas the observed crevasses have been evolving for decades. We added these two points on page 11, line 4-6.

*page 2*

*l.6 suggested rewrite: "The calving of icebergs is difficult to model because the processes involved, such as the initiation, propagation, and orientation of crevasses are not well*

*understood, and direct observations are rare...”*

Agreed. The sentence has been rewritten as suggested on page 2, line 7-9.

*l.20 "In this study, however, ...” I do not quite understand this sentence.*

The sentence is rewritten to "In their study, however, the crevasse propagation process is not modeled. The crevasses were either zero in size or propagating through the entire ice thickness to create a calving event.” at page 2, line 21-23.

*l.24 rewrite: "The model simulations are conducted ...”*

Done on page 2, line 27.

*page 5*

*l.13 "matches the observed surface velocity”*

Done on page 5, line 18.

*l.15 "Here, the simulation time is short, ...”*

Done on page 5, line 20.

*l.26 "For SSA and HO, zb is known because ...”*

Done on page 6, line 1.

*page 6*

*l.6 "In order to obtain a realistic ...”*

Done on page 6, line 10.

*l.18 a textbook reference for the three modes of fracture may be worthwhile*

Agreed. We added a textbook reference "Mechanics: Fundamentals and Applications, Third Edition” by T. L. Anderson on page 6, line 21 (Anderson, 2005).

*l.25 the authors fixed the typo in eq (20), but it appears that the following text has not been updated. Presumably it should read something like, "$\sigma_{xx}$' $= \sigma_{xx} + p$, is the deviatoric stress, with $\sigma_{xx}$ being the longitudinal stress and p ..."*

That is correct. Thank you for catching this. The manuscript has been modified accordingly on page 6, line 29.

*page 7*

*l.20 I would suggest moving the requirement of minimum depth being smaller than 1 m up to line 3 of the page (specifying under what conditions the $K > K_c$ criterion actually leads to crevasse propagation).*

Agreed. The statement has been moved up to page 7, line 6-8.

*l.25 "20 m width to 60-70m..."*

Done on page 7, line 29.

*l.32 There seems to be something wrong with this sentence.*

The sentence has been rewritten to "To validate our FS modeling of the grounding line dynamics, we run the Experiment 3 of MISMIP. In this experiment, we model the grounding line migration resulting from a change in ice rheology on an over-deepened bed." Please see page 8, line 5-8.

*page 8*

*l.22 "The initial crevasse positions ..."*

Done on page 8, line 27-28.

*l.26 "In the fourth set of experiments ..."*

Done on page 8, line 32.

*page 9*

*l.18 I would suggest renaming this subsection maybe "Deviation from hydrostatic equilibrium"*

Agreed. Done on page 9, line 21.

*l.30 "... downstream of the grounding line."*

Done on page 10, line 3.

*page 10*

*l.1 It should be stated that the crevasse stops growing when K falls below Kc.*

Correct. Done on page 10, line 4.

*l.26 "...stops growing."*

Done on page 10, line 29.

*page 11*

*l.16 "Over time, the stress at ..."*

Done on page 11, line 21.

*page 12*

*l.11 This statement appears too strong. Suggested rewrite: "This conclusion may help reconcile the previous studies ..."*

Agreed. Done on page 12, line 15-16.

*l.17/18 the revised sentence needs rewriting*

The sentence has been rewritten "It would be of interest to generalize the present simulation to a 3D geometry with the inclusion of multiple crevasses and a moving ice front." and "It would also be useful to include in the current model the subcritical propagation or the damage mechanics, which is needed to improve the simulation of the initial propagation from an infinitesimal crevasse to a ∼1 m deep, ∼10 m wide crevasse." at page 12, line 22-23 and line 26-29.

*l.23 "With these additions ..."*

Done on page 12, line 30.

*l.26 I'd suggest spelling out Thwaites Glacier here (rather than TG).*

Agreed. Done on page 12, line 32-33.

*l.30 "... that assume hydrostatic equilibrium ..."*

Done on page 13, line 3.

*Fig 5: The colors between panels (a) and (b) don't match. The blue profiles (12 and 13) have the grounding line retreat back to x~700 km. In panel (b) the grounding line profiles appear as if they increase monotonically (i.e., not showing the hysteresis loop).*

The colors in Fig.5 has been changed to show the hysteresis loop more clearly.

*Fig 7, caption: Which experiment is the modeled data is from? Labels of horizontal and vertical axes should be consistent (i.e., specify or leave out "surface"/"bottom" on both).*

Agreed. The modeled data is from Exp. A. This information has been added in the caption. The labels have been modified to have "surface/bottom" on both axes.

*Fig 8: It is difficult to distinguish between the red and the magenta curves (maybe use green or some other color instead). If I read this figure correctly, crevasse propagation can happen only if a given curve goes through the bottom right quadrant? Maybe this can be highlighted visually, with shading or a box?*

The magenta curves have been changed to green. A gray box has been added in the fourth quadrant to highlight the conditions under which crevasses propagate.

**2 Response to Reviewer Jeremy Bassis**

Detailed below are our point-by-point responses to the comments from reviewer Jeremy Bassis. Reviewer's comments are in blue, followed by our responses in black.

*Overall, I think the authors have done a commendable job of addressing reviewer comments. In addition to a few technical corrections, I still have two questions/comments, but these may reflect my own misunderstanding of the methods used by the authors and I leave it at the discretion of the editors and authors whether these comments need to be addressed prior to publication. I also strongly suggest that the authors provide inputs to their model*

*as supplementary data tables or figures. I think most of the information is included as figures, but I dont think the authors provide the mass balance field they use and this should be included as a figure or as a supplementary table. Personally, I would advocate that the authors include all inputs (basal friction, mass balance, bed geometry, etc) as a data table supplementary information to allow others to more easily reproduce the authors results. My major comments refer to previous comments and are labeled as such.*

We added a netcdf file in the supplementary material to supply the inputs of our model: latitude, longitude, geometry, surface mass balance and friction coefficient.

*Response to response to comment 1: I'm still slightly concerned about the distinction between deviatoric and resistive stresses which is explored in the response to my comment from the previous review. I've pondered this and I think I'm missing something here. As I understand it, the authors substitute deviatoric stresses into Van der Veen's resistive stress formulation to calculate stress intensities. This is justified based on the fact that the pressure only deviates by a small amount from the hydrostatic pressure (cryostatic, lithostatic, what ever). The definition of resistive stresses given by the authors in the reply appears to diverge from Van der Veen's. Van der Veen defines $R_{xx} = \sigma_{xx} + \rho g(h_s - z)$ not $R_{xx} = \sigma_{xx} + \rho g H$, where hs denotes the surface elevation and h is the ice thickness. I assume this is a typo in the authors reply (?). Nonetheless, my understanding is that the authors argue that deviatoric stress $\sigma'_{xx} = \sigma_{xx} + p$ is (nearly) equivalent to resistive stresses when pressure p is (nearly) hydrostatic, i.e., $p = \rho g(h_s - z)$ and the authors argue that pressure is in fact within a few percent of hydrostatic. However, in the shelfy-stream approximation, the vertical force balance condition requires that $p = \rho g(h - z) - \sigma'_{xx} + \sigma'_{yy}$. At least in the shallow approximation, the pressure is reduced by horizontal stretching. In the interior of the ice sheet, where the ice is frozen to its bed, longitudinal stretching is small and pressure is approximately hydrostatic. A consequence is that $2\sigma'_{xx} = R_{xx}$ (again in the shallow shelf limit). This argument only holds for the shallow models and doesn't hold for the full Stokes calculation, but I would expect qualitatively similar results in which deviatoric stresses act to reduce the pressure. This leaves me confused as to what the authors did to calculate crevasse depths and why the pressure in their model is (nearly) hydrostatic. I assume the authors did the right thing, but it is hard to decipher what that thing is.*

The argument in our reply that $R_{xx} = \sigma_{xx} + \rho g H$ is a typo, since this is only valid at the ice bottom (also see our response to Reviewer 1). At each depth, we agree that $R_{xx} = \sigma_{xx} + \rho g(h_s - z)$. We also agree that the horizontal stretching will act to reduce the pressure. Immediately after the crevasse propagates, there is a singularity in the pressure field. We find, however, that after adjusting viscously, the deviation from the hydrostatic pressure is of second order magnitude compare to the hydrostatic pressure. The deviatoric stress is also used to computed the stress intensity factor in Krug et al. (2014). Finally, we

conducted an experiment where we use the hydrostatic pressure instead of the computed pressure and we obtained similar results. This is clarified in the manuscript at page 7, line 14-15.

*Response to response to comment 2: This is probably reasonable, but again Im still slightly confused. If crevasses have initial width w0, then the rate at which they widen initially will be proportional to the extensional strain rate, say $dw/dt = E_{xx}w$, where $E_{xx}$ is a measure of the extensional strain rate opening the crevasse. This results in an exponential widening rate early on in the evolution of the crevasse. This suggests that if crevasses are initially 20 m wide and the width triples to 60 m in time interval $\Delta t$, then a 1 cm wide (my calculations suggest more like 4-6 mm) should widen to 3 cm in $\Delta t$ just based on the kinematics of the flow field. This would suggest that the width of crevasses later on in the simulation may be a function of the initial width of the crevasses and better (or worse) agreement with observations could be obtained merely by adjusting the initial width of the crevasse. Again, I assume the authors have done grid sensitivity experiments, but a statement or two (or a plot) to point out that the width of crevasses throughout the simulation is only weakly dependent on numerical resolution would be useful to fortify this in the audiences mind.*

We agree that the widening rate of a crevasse is proportional to the width itself at the initial elastic opening stage. Yet we do not attempt to model the initial widening process, which needs cm level mesh resolution and is too computationally demanding. We assume that the crevasse grows to a width at the order of 10 m immediately and then we model the widening of the crevasse in response to the viscous flow of ice, not the elastic deformation. In this process, the widening of the crevasse is mostly dependent on the extensional strain rate, not on the width of the crevasse. We have conducted two experiments with an initial width of 10 m and 40 m and at the end of the simulations, we find that the differences in width of the crevasse is only in the order of ∼10 m. We clarified this issue on the modeling of the crevasse on page 7, line 30-32.

*Technical corrections*

*I still have questions about the role of damping in simulating the stress field (especially as it translates to crevasse depths). The damping term can create an additional stress that acts to open crevasse because the ice shelf is never exactly in hydrostatic equilibrium with the ocean. However, this may be a higher-order numerical question.*

We agree that this dampening term may (or may not) create an additional stress that might impact the crevasse propagation. Numerically, adding the dampening term is however a necessity and a standard procedure for FS solutions. Without this dampening term, the system cannot reach a stable and reasonable solution. The manuscript is not changed for this comment.

*Page 7, line 10: The time scale of major calving events from ice shelves is years to decades, not days to weeks. The days to weeks time frame is appropriate for grounded Greenland glaciers.*

Agreed. The statement has been removed from the manuscript at page 7, line 23-24.

[revised manuscript text omitted]